



# Measurement Report: Atmospheric CH4 at regional stations of the Korea Meteorological Administration/ Global Atmosphere Watch Programme: measurement, characteristics and long-term changes of its drivers

Haeyoung Lee[*], Won-Ick Seo, Shanlan Li, Soojeong Lee, Samuel Kenea, and Sangwon Joo

National Institute of Meteorological Sciences, Jeju, 63568, Republic of Korea

*Correspondence to Haeyoung Lee (leehy80@korea.kr)*

Abstract. To quantify $CH_4$ emissions at policy-relevant spatial scales, the Korea Meteorological Administration (KMA) started monitoring its atmospheric levels in 1999 at Anmyeondo (AMY), and expanded monitoring to Jeju Gosan Suwolbong (JGS) and Ulleungdo (ULD) in 2012. The monitoring system consists of a Cavity Ring Down Spectrometer (CRDS) and a new cryogenic drying method, with a measurement uncertainty (68% c.i.) of 0.7–0.8 ppb. To determine the regional characteristics of $CH_4$ at each

KMA station, we assessed the $CH_4$ level relative to local background ($CH_4xs$), analyzed local surface winds and $CH_4$ with bivariate polar plots, and investigated $CH_4$ diurnal cycles. We also compared the $CH_4$ levels measured at KMA stations with those measured at the Mt. Waliguan (WLG) station in China and Ryori (ROY) station in Japan. $CH_4xs$ followed the order AMY (55.3±37.7 ppb) > JGS (24.1±10.2 ppb) > ULD (7.4±3.9 ppb). Although $CH_4$ was observed in well mixed air at AMY, it was higher than at other KMA stations, indicating that it was affected not only by local sources but also by distant air masses. Annual mean $CH_4$ was

highest at AMY among all East Asian stations, while its seasonal amplitude was smaller than at JGS, which was strongly affected in the summer by local biogenic activities. From the long-term records at AMY, we confirmed that the source of $CH_4xs$ changed from the past (2006 to 2010) to recent (2016 to 2020) years in East Asia. Especially in northern China, $CH_4xs$ was mainly attributed to fossil fuel combustion or biomass burning during 2006–2010, but mainly to biogenic activities during 2016–2020, as indicated by decreasing $\delta^{13}CH_4$ from the northern China. $CH_4$ emissions in the southern part of China and in South Korea were enhanced

by biogenic signals during 2016-2020. We confirmed that long-term high-quality data can help understand changes in $CH_4$ emissions in East Asia.

## 1. Introduction

Atmospheric methane ($CH_4$) is an important greenhouse gas and is one of the main drivers of climate change on Earth. The global atmospheric $CH_4$ abundance was 1889±2 ppb in 2020, increasing 2.6 times since 1750 (~722 ppb, pre-industrial period); the

relative $CH_4$ increase since pre-industrial is greater than other major greenhouse gases such as $CO_2$ (1.5 times) and nitrous oxide (1.2 times) (WMO, 2021). Recently, $CH_4$ has gained substantial interest because of its relatively shorter lifetime in the atmosphere (~ 9 year) compared with that of other long-lived greenhouse gases (Prinn et al., 2005). $CH_4$ emission reduction may thus be an





effective method to partially mitigate climate change. The 6th Intergovernmental Panel on Climate Change (IPCC) reported that, if strong and sustained $CH_4$ emission reductions are integrated with air pollution controls, net warming could decrease in the long term because of the short lifetime of both $CH_4$ and aerosols (IPCC, 2021).

To reduce the atmospheric $CH_4$ burden, its emissions and sinks must first be quantified. $CH_4$ loss is primarily attributed to reaction
with hydroxyl radicals (OH), which are part of atmospheric photochemical cycles, while there are various natural (wetlands, freshwaters, and geological) and anthropogenic $CH_4$ sources (agriculture, waste, fossil fuels, and biomass burning) with different spatial and temporal distributions (Jackson et al., 2020, Lan et al., 2021). Because of its diverse sources in different regions, high resolution, quality data can help quantify the atmospheric $CH_4$ budget.

In the World Meteorological Organization (WMO), Global Atmosphere Watch Programme (GAW), there are 170 stations that
monitor atmospheric $CH_4$, but with poor spatial coverage in Asia (gawsis.meteoswiss.ch, last access: November 2021).

Among $CH_4$ sources, rice agriculture is intense in Asia, mainly in China and India (Kai et al., 2011). China has also the largest anthropogenic $CH_4$ emissions in the world (Janssens-Maenhout et al., 2019).South Korea ranks among the world's top three importers of liquefied natural gas (LNG), following Japan and China (eia.gov/international/analysis/country/KOR. last access: November 2021). In this regard, the Korean atmospheric monitoring network is important to understand not only South Korea $CH_4$
burden but also Asia continent because of its location, as it is sensitive to air masses transported from Asia, and especially from China.

South Korea's atmospheric $CH_4$ monitoring history started at Tae-Ahn Peninsula (TAP, 36.74°N, 126.13°E, 20 m above sea level) by Korea Centre for Atmospheric Environment Research in the western part of Korea in 1990, with weekly flask-air sample collection as a part of the U.S. National Oceanic and Atmospheric Administration (NOAA), Global Monitoring Laboratory (GML),
Cooperative Global Air Sampling Network (http://www.esrl.noaa.gov/gmd/ccgg/flask.php). Since 1999, the Korea Meteorological Administration (KMA) has been monitoring atmospheric $CH_4$ with quasi-continuous measurements at Anmyeondo (AMY, 36.53°N, 126.32°E, a 40 m tower whose base is 46 m above sea level), approximately 28 km from TAP. In 2012, KMA expanded its monitoring network to capture data from the south-west (Jeju Gosan Suwolbong, JGS, 33.30°N, 126.16°E) and east (Ulleungdo, ULD, 37.48°N, 130.90°E) of Korea to cover the entire peninsula for a better understanding of $CH_4$ sources and its characteristics.
However, there is no published description of measurement quality, regional characteristics and long-term trends of $CH_4$ for Korea network.

A few studies reported that $CH_4$ levels are affected by emissions from Russian wetlands and local rice cultivation near TAP (Dlugokencky et al., 1993; Kim et al., 2014). In 2019, observations at AMY indicate larger emissions compared with previous years, which were caused by soil temperature and moisture changes (Kenea et al., 2021). In summer, high atmospheric $CH_4$ levels
were observed in airborne measurements because of biogenic sources such as rice paddies, landfills, and livestock (Li et al., 2020). A large ratio, $CH_4/C_2H_6$, was observed during KORUS-AQ campaign from May to June 2016 in the vicinity of industrial regions in west coast of South Korea (Li et al., 2022). These findings are related to the case studies so that it can be difficult to figure out the representative long-term and regional characteristics in Korea. Also even if $CH_4$ is monitored at regional scale during long-term period, poor measurement quality can lead to misinterpretation of the $CH_4$ budget, preventing development of science-based
policies. Additionally, both measurement uncertainty and inadequate assessment of background air can limit the accuracy of observation-based estimates for local or regional scale greenhouse gas emissions (Graven et al., 2012; Turnbull et al., 2009, 2015; Lee et al., 2019).

In this paper, we present $CH_4$ data quality procedures and processing methods at three KMA monitoring stations, including measurement uncertainties. We analyzed the characteristics of $CH_4$ at the KMA stations from 2016 to 2020 and compared the data

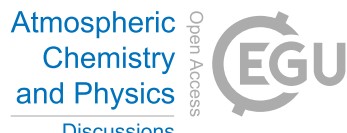

with those collected at other stations in East Asia: the global background WMO/GAW station in Waliguan (WLG, 36.28°N, 100.90°E, 3810 m), China; and the WMO/GAW station at Ryori (RYO, 39.03°N, 141.82°E, 260 m) in Japan, which reflects the global growth rate (Watanabe et al., 2000). In addition, we investigated the changes in $CH_4$ enhancement from 1999 to 2021 and analyzed the source regions based on measurements of $\delta C_{(CH4)}$ in flask-air samples to trace the major source changes. Furthermore, this study can serve as a reference for KMA data archived at the World Data Centre for Greenhouse Gases.

## 2.    Experiment

### 2.1    Sampling sites

The locations of AMY, JGS, and ULD stations are shown in Fig. 1 and summaries of the measurement systems are in Table 1. Detailed information was provided in Lee et al. (2019). Only key information focusing on $CH_4$ is summarized here.

AMY is located in the western part of Korea approximately 28 km south from TAP and 130 km southwest from the megacity of Seoul. Within 50 km of AMY, the second largest rice paddies and largest livestock industry of Korea are present. The largest coal and heavy oil fired thermal power plants in Korea are present within 35 km of this station, to the north-east and south-east, respectively, and the largest LNG power plant in Korea is 100 km to the north east of this station. The local region mainly consists of agricultural land growing rice, sweet potatoes, and onions, and the area is also known for its leisure opportunities during summer. The west and south sides of AMY are open to the sea, with a large tidal mudflat with many pine trees along the coast.

JGS is located in the western part of Jeju Island, which is the largest volcanic island (1,845.88 km²) in the south-west of Korea and is approximately 90 km from the mainland. The major industries here are tourism and livestock, which focuses on horses and pigs. JGS is located within a famous Global Geo-park that has outcrops of volcanic deposits exposed along the coastal cliff. Next to JGS, agriculture is widespread, with potatoes, garlic, and onions being the main crops in the largest plain in Jeju Island. The station is open to the sea from the south-west to north-west, with the cliffs comprising volcanic basalt rocks. The sea to the south is connected to the East China Sea and the sea to the west is linked to the Yellow Sea.

ULD is located in the east of Ulleung Island, which is in the eastern part of Korea and approximately 155 km from the mainland. In the south-eastern part of the Korea Peninsula, numerous steel, chemical, and petrochemical industries are present along the coastline, within approximately 200–250 km from the island. There are two large natural gas power plants. Ulleung Island covers 72 km², and has a volcanic origin, being a rocky steep-sided island that is the top of a large stratovolcano that has a maximum elevation of 984 m. This peak is located northwest of ULD. There are a few small mountains with heights from 500 to 960 m a.s.l., within 5 km to the north and southeast of the station. Because of those geological features, ULD is mainly affected by airflow from over the hill to the southwest and by downslope winds from northeast. Before 2016, there was a garbage incineration plant 300 m from ULD. In the southwestern area, there is a small brickyard 200 m from the station and a garbage incinerator within 100 m. The garbage incineration facility was moved to the north side of island in December 2016. Therefore, many studies do not include the data before 2017. Farming and fishing industries are very active on the island, although there are no farms in the southern area.

An automatic weather station (AWS) was installed at AMY near the air sampling inlet, and 10 m above the station at JGS and ULD, independent from the air inlet tower.





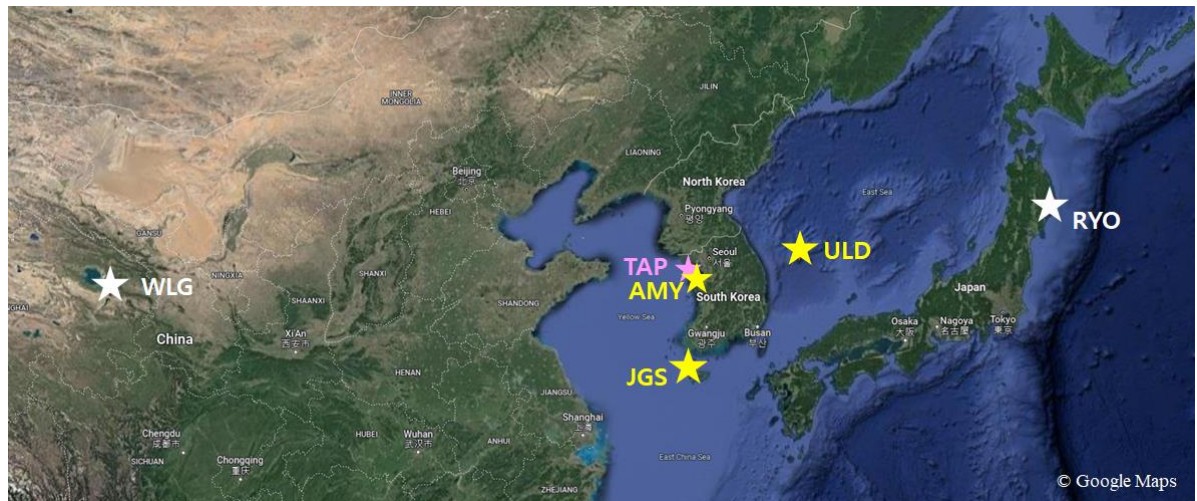

Figure 1. Locations of KMA CH$_4$ monitoring stations in Korea: Anmyeondo (AMY), Jejudo Gosan Suwolbong (JGS) and Ullengdo (ULD). Tae-an Peninsula (TAP, 36.74°N, 126.13°E), part of NOAA's flask-air sampling network, is 28 km from AMY in South Korea. Mt. Waliguan (WLG, 36.28°N, 100.90°E) and Ryori (RYO, 39.03°N, 141.82°E) are located in China and Japan, respectively. This map is derived from google map.

## 2.2    Measurement environment and instrument

At all three stations, the measurement system consists of 1) inlet, 2) pump, 3) drying system, and 4) analyzer. Detailed information of the system was discussed by Lee et al. (2019).

1) Inlet: Dekabon sampling tubing (Nitta Moore 1300-10, I.D. 6.8 mm, O.D. 10 mm, high-density polyethylene jacket, overlapped aluminum tape, and ethylene copolymer liner) with a stainless-steel filter (D 4.7 cm, pore size 5 µm) mounted on a plastic mesh holder is installed on the intake and connected to the pump. The inlet height was changed at AMY in 2004 and at JGS in 2017 (Table 1).

2) Pump: A KNF diaphragm pump (N145.1.2AN.18, Germany, 55 L/min, 7 bar in AMY; N035AN.18, Germany, 30 L/min, 4 bar in JGS and ULD) is installed between the inlet and drying system.

3) Drying system: Sample air is dried with a cryogenic method (CT-90, Operon, Korea, –90°C). Inside the drying system, there are two chambers with two steps; ambient air is cooled to –20°C in the first chamber, and then to –50°C in the second chamber. This system was installed in 2012 at all three KMA stations.

4) Analyzer: A model G2301 (Picarro, USA) was installed in October 2011, and it became the official CH$_4$ measurement system at AMY starting February 1, 2016. Before February 2016 (G2301), a GC-FID was used to monitor atmospheric CH$_4$. CRDS records atmospheric CH$_4$ every 5 s across the KMA Greenhouse Gas (GHG) network while the GC-FID measured CH$_4$ every 30 min. At JGS, the monitoring of atmospheric CH$_4$ started with the use of G1301 in 2012, which was changed to G2401 from 2020. G2401 has been used from 2012 at ULD.





Table 1. Information on the three KMA CH$_4$ monitoring stations in Korea

| Station (ID) | Longitude Latitude Altitude | Inlet height (period) | Instrument Model (period) | Drying method (period) | Standard scale (period) |
|---|---|---|---|---|---|
| Anmyeondo (AMY) | 126.32°E 36.53°N 47 m a.s.l | 20 m (1999 to 2004) | GC-FID (1999 to 2016 Feb) | Three step dehumidification system 1) –4°C cold trap 2) Nafion™ 3) Mg(ClO$_4$)$_2$ (1999 to 2011) | KRISS (1999 to 2011) |
| | | 40 m (since 2004) | CRDS 2301 for CO$_2$ and CH$_4$ (2016 Feb to present) | Cryogenic system (since 2012) | WMO-X2004A (2012 to present) |
| Jejudo Gosan Suwolbong (JGS) | 126.16°E 33.30°N 71.47 m a. s.l | 6 m (2012 to 2017) 12 m (since 2017) | CRDS 1301 for CO$_2$ and CH$_4$ (2012 to 2019) CRDS 2401 for CO$_2$, CH$_4$, and CO (since 2020) | Cryogenic system | WMO-X2004A (2012 to present) |
| Ulleungdo* (ULD) | 130.90°E 37.48°N 220.9 m a.s.l | 10 m (since 2012) | CRDS 2401 for CO$_2$, CH$_4$, and CO (since 2012) | Cryogenic system | WMO-X2004A (2012 to present) |

*ULD is not registered in the GAW network.

**2.3      Calibration method**

Our highest level standards are designated "laboratory standards". We have four laboratory standards prepared by the WMO GAW Central Calibration Laboratory (CCL) on the WMO-X2004A scale in the range 1700 to 2500 ppb with uncertainties of less than 2 ppb (95% confidence level, coverage factor k=2, gml.noaa.gov/ccl/ch4).

For working standards, dry, ambient air is compressed into a cylinder at AMY with a range of roughly 1800-2500 ppb. To bracket

the measurement range, we diluted collected air to around 1700 ppb with pure zero air.

Filled working standards are sent to a central laboratory of the National Institute of Meteorological Sciences (NIMS) in Jeju for calibration against the laboratory standards. The scale was transferred with a CRDS (G2401, Picarro, USA). The scale propagation uncertainty is described in section 3.1. Normally the difference in H$_2$O between laboratory and working standards measured by CRDS is ~0.00054%, which leads to a bias of 0.01 to 0.014 ppb for CH$_4$ in the given range according to the Eq. (1) from Rella at

al.(2013). This value is negligible so it was not considered as a factor for the propagated uncertainty (Eq. 1)

$$\frac{C_{dilution}}{C_{dry}} = 1 - 0.01 H_{act} \qquad \text{(Eq. 1)}$$

where $C$ is the CH$_4$ mole fraction and $H_{act}$ is the actual water mole fraction (in %). Here $H_{act}$ is the H$_2$O difference between standard

gases and samples.



Analyzer response had been calibrated every two weeks for all stations before 2019 Dec. but it was changed to 5 or 6 days with different calibration frequency at each station based on the reproducibility; all 4 working standard gases with a range of 1700-2500 ppb at intervals of 200-300 ppb were measured by the CRDS for 40–50 min. Only the last 10 min of data were used for the calibration of $CH_4$ to ensure instrument stability (Lee et al., 2021). Our ability to maintain and propagate the WMO-2004A scale

was shown through the $6^{th}$ Round Robin comparison test (RR) of standards hosted by the CCL (https://www.esrl.noaa.gov/gmd/ccgg/wmorr/wmorr_results.php, the difference for low $CH_4$ levels was 0.7±0.7 ppb while that for high $CH_4$ levels was $0.6 \pm 0.7$ ppb)

When we started monitoring atmospheric $CH_4$ at AMY in 1999, the GC-FID response was calibrated every 1.5 h with a 1-point calibration against the KRISS scale until February 2016. During this period, we used standards that were certified directly by

KRISS without working standards. KRISS and WMO-2004A scales agreed well with a difference from -0.1 to 0.8 ppb (gml.noaa.gov/webdata/ccgg/wmorr/rr5, last access: Jun 2022). WMO round robin comparison of standard scales (Round robin number 5) included measurements with our GC-FID; differences with the CCL were from –0.3 to 1.3 ppb in the range 1756 to 1819 ppb.

**2.4      Data quality control/assurance process and baseline selection method**

### 2.4.1   Auto and manual quality control/assurance process

All data were collected and stored at NIMS in Jeju, Korea. Raw data at 5 s intervals (L0 data) were processed as L1 data with 1) auto flagging and 2) manual flagging. Auto flagging involved 6 criteria, including instrument malfunction, instrument detection limit, and value outside the given calibration range. Manual flags were assigned by technicians at each station according to the

logbook based on inlet filter exchange, diaphragm pump error, low flow rate, dehumidification system error, calibration periods, experimental periods such as participation in comparison experiments, observatory environmental issue such as construction next to a station, extreme weather, or other issues related to the instrument. These codes refer to definitions by the World Data Centre for reactive gases and aerosols maintained by EBAS for the GAW Programme (http://www.nilu.no/projects/ccc/flags/flags.html, last access: 23 Aug. 2022) that were modified for the Korea network. Data with flags were reviewed by scientists at NIMS, and

only valid data were averaged into Level 2 (L2) hourly average data. To define valid data, all data were compared between South Korea stations and other global stations at similar latitude to Korea.

One of the way of quality assurance, a co-located comparison of discrete samples collected at AMY, the sampled flasks were analyzed by NOAA/GML and compared with our in-situ analyzer results. This comparison between L2 hourly data from the CRDS and weekly flask-air samples collected at AMY has been ongoing since December 2013. The mean difference, flask minus CRDS

hourly mean in situ, was $2.2 \pm 11.8$ ppb from 2016 to 2020, which is close to GAW's compatibility goal for $CH_4$ ($\pm 2$ ppb) (Fig. S1). During the period of GC-FID measurements, the average difference ($\pm 1$ SD) between KMA and NOAA flasks was $5.2 \pm 15.6$ ppb, which is greater than the difference since CRDS observations started but reasonable as per the GAW extended compatibility goal of $\pm 5$ ppb.





### 2.4.2 Regional background selection method

To understand the atmospheric CH$_4$ measurements and CH$_4$ growth rate, data representing well-mixed air should be selected for analysis on a regional scale. There are many methods to select data for baseline such as using related tracers, AWS data, or statistical methods (Fang et al., 2015; Chambers et al., 2016; Bacastow et al., 1985; Lowe et al., 1979). For Korea network, we
used a statistical method and it was described in detail by Seo et al.(2021). There are three criteria; 1) hourly standard deviation (HS), 2) differences in consecutive hourly values (CD) and standard deviation of a 30-day moving average (MS). After applying HS and CD to observed data, the data were defined as L3 hourly data. Even though the data were selected by HS and CD, high CH$_4$ levels remained because of long-lasting stagnant conditions (e.g. over 6 days). Therefore, we also apply MS (Table 2). MS was multiplied by α, which was determined empirically. This process retained 21–52% of the data at each station, which were
defined as L3 hourly and daily data based on observations. The method developed by Thoning et al. (1989) was used to fit smooth curves to L3 daily averages. These curves reduce noise induced by synoptic-scale atmospheric variability, fill measurement gaps, and are used to represent the regional baseline. Finally, we can get the L3 monthly data, long-term trend and seasonal amplitude, which indicates the magnitude of the peak to trough of the detrended seasonal cycle, after applying Thoning et al. (1989).

Table 2. Criteria and percentage of selected background levels from observed data at each station. HS: Hourly standard deviation, CD: consecutive hourly value, MS: standard deviation of a 30-day moving average

| Station ID | AMY | JGS | ULD |
|---|---|---|---|
| Data period | 1999 to 2020 | 2012 to 2020 | 2014 to 2020 |
| HS [ppb] | 2.1 | 2.1 | 2.8 |
| CD [ppb] | 4.9 | 5.2 | 3.6 |
| MS [ppb] | | $1.8\sigma_{30d}$ | |
| Spring, MAM [%] | 29.1 | 46.6 | 57.9 |
| Summer, JJA [%] | 11.0 | 33.5 | 37.6 |
| Autumn, SON [%] | 16.9 | 30.9 | 53.2 |
| Winter, DJF [%] | 28.4 | 49.1 | 58.9 |
| Total [%] | 21.3 | 40.64 | 52.2 |

CH$_4$ data were produced from 2016 under the same conditions for all three stations; however, ULD was affected by emissions
from a garbage incinerator until Dec. 2016, while AMY was affected by a malfunction of the drying system for 26 Aug-9 Sep. 2016. The garbage incinerator was moved to the northeast part of the island in December 2016. Therefore, we compared data from the three stations from 2016 to 2020, excluding the periods mentioned above.

In section 3.5, for comparison of our station annual/monthly mean and seasonal amplitude to those parameters calculated from
other Asian stations (Mt. Waliguan global GAW station (WLG, 100.90°E, 36.28° N, 3810 m) and Ryori regional GAW station (RYO, 141.82° E, 39.03° N, 260 m)), we downloaded daily data for these stations from the World Data Centre for Greenhouse





Gases (http://gaw.kishou.go.jp). We applied the Thoning et al. (1989) method to each daily data set to get annual/monthly mean and seasonal amplitude to compare.

### 2.5    Flask-air data

Long-term data on $CH_4$ and its isotopes ($\delta^{13}C$ in $CH_4$, hereafter $\delta^{13}C_{(CH4)}$) were collected at TAP, 24 km away from AMY. Samples were collected weekly between 1200 and 1800 (Korea Local Time), when boundary layer height (BLH) was maximum. The collected samples were sent to Boulder, Colorado for measurement of $CH_4$ at NOAA and to INSTAAR (Institute of Arctic and Alpine Research, University of Colorado) for $\delta^{13}C_{(CH4)}$ analysis. Samples were analyzed from 1990 for $CH_4$ and from 2000 for $\delta^{13}C_{(CH4)}$. Since TAP and AMY are only 24 km apart, their data are representative of the same region under large synoptic

conditions, especially for well mixed air. These data were thus used to trace the changes in the surrounding environment in East Asia (section 3.5). AMY started sampling in December 2013 and these data were used only for characterization of $CH_4$ at AMY in section 3.5.

### 2.6    Hybrid Single-Particle Lagrangian Integrated Trajectory model (HYSPLIT) cluster analysis

We downloaded and installed the HYSTPLT with window version and used the built-in algorithm. HYSPLIT trajectories were calculated using the Global Data Assimilation and Prediction System (GDAPS) at a horizontal resolution of 25 km to determine the origin of air masses transported to TAP during 2005–2020. The back trajectories were calculated for 96 h periods at 3 h intervals, with 500 m altitude matching the time of each flask-air sample. Based on a cluster analysis, northern China (CN) accounted for 27% of all air masses; these originated in Russia and travelled through Mongolia and northeast China. Southern China (CS)

accounted for 6%; these air masses originated from East China sea and the southern part of China. Air masses from Korea local (KL) reflected emissions from the Korean Peninsula and Japan, accounting for 17% of the total air masses. Other sectors were also analyzed, but there were no significant or representative changes in emissions, so they are not reported herein.

### 2.7    Potential Source Strength (PSS) analysis

We calculated PSS using the trajectory statistics approach, which has often been applied to estimate the potential source areas of air pollutants (Remann et al., 2004, 2008; Li et al., 2017). The trajectory statistics approach calculates air mass residence time weighted mean mole fractions for target compounds ($CH_4$ in this study) for the domain with $0.5° \times 0.5°$ grids using the following formula (Eq. 2):


$$\overline{C_{(i,j)}} = \frac{\sum_{a=1}^{M} T_{(i,j,a)} C_a}{\sum_{a=1}^{M} T_{(i,j,a)}} \qquad (Eq. 2)$$

where C(i, j) represents the potential source strength of the cell i, j as a potential source region of the target compound ($CH_4$); a is the index of the trajectory; M is the total number of trajectories that passed through cell i, j; Ca is the enhanced mole fraction (difference from background mole fractions mentioned in section 3.2 below) measured during the arrival of trajectory a; and Ti,j,a





is the residence time of trajectory a spent over grid cell i, j. Residence times were calculated using the method described by Poirot and Wishinski (1998). To consider the influence of air masses on emissions at ground level, the air masses passing above the BLH were excluded. BLH was obtained from the HYSPLIT model. To exclude the influences of emission sources surrounding AMY, enhanced $CH_4$ data with wind speeds lower than 2 m/s were excluded from the PSS analysis.

## 3.    Results and Discussion

### 3.1    Measurement uncertainty

Observed $CH_4$ is influenced by natural atmospheric variability and measurement procedures. Natural atmospheric variability can be represented as the standard deviation of all measurements contributing to a time-average, after the contribution of experimental noise is accounted for. The measurement uncertainty is critical to provide information on data quality so that users can understand

the limitations and reliability of values. According to previous studies, the total measurement uncertainty consists of multiple uncertainty components (Andrews et al., 2014, Verhulst et al., 2017). For the KMA network, a measurement uncertainty of approximately 0.11 ppm has been calculated for $CO_2$ with limited but practical components (Lee et al., 2019). Using the same method used for $CO_2$, we calculated a practical realistic measurement uncertainty for $CH_4$ in the KMA network (Eq.3). Based on the measurement of target cylinders and a co-located comparison of measurements at AMY and JGS, we assumed systematic

biases to be negligible (http://empa.ch/web/s503/wcc-empa, last access: January 2022).

$$(U_T)^2 = (U_{h2o})^2 + (U_P)^2 + (U_r)^2 + (U_{scale})^2 \tag{Eq. 3}$$

where $U_T$ is the total measurement uncertainty in the reported dry-air mole fractions; $U_{h2o}$ is the uncertainty from the drying system;
$U_p$ is repeatability; $U_r$ is reproducibility; and $U_{scale}$ is the uncertainty of propagating the WMO-X2004A $CH_4$ scale to working standard gases.

Since working standards had nearly the same level of $H_2O$ as laboratory standards through the CRDS, we only considered the $CH_4$ dilution offsets between standards and sample air while estimating the uncertainty. $U_{h2o}$ was computed from the differences in $H_2O$
(%) between the ambient airstream through the drying system and standard gases injected directly, bypassing the drying system. An ideal measurement would be through the analysis of the standard gases and air samples after they pass through the same drying system (WMO, 2016). However, our drying efficiency was not constant, so we injected standard gases directly as a reference value of $H_2O$. We defined hourly mean $H_2O$ values from standard gases in the calibration period as a reference value. This value was applied to ambient air to calculate $U_{h2o}$ before the next calibration. This meant that the uncertainty component was time dependent.
Eq.(1) was applied to this factor, where $H_{act}$ was the difference between $H_2O$ in samples and standard gases. Hourly $CH_4$ dilution maximum offsets are up to 0.009 ppb at AMY, 0.006 ppb at JGS and 0.009 ppb at ULD from 2016 to 2020. Since positive and negative values were found, we used the following equation (Eq. 4):

$$U_x = \sqrt{\frac{\sum_{i=1}^{N}(x_i)^2}{N}} \tag{Eq. 4}$$






where $U_x$ represents $U_{h2o}$; x is the hourly $CH_4$ dilution offsets from Eq(1); N is the total number of hourly mean values. $U_{h2o}$ is tabulated for each station in table 3. This uncertainty term was smallest (0.006 to 0.008 ppb) among all uncertainty factors in the KMA network.

We expressed $U_r$ as the standard deviation of all drift during the experimental period using Eq (4), where $U_x$ represents $U_r$; $x_i$ is the drift occurring between calibration episodes; and N is the total number of data. They are tabulated with other uncertainty terms by site in table 3. We determined $U_r$ as the differences in $CH_4$ measured from cylinders with subsequent calibrations after two weeks. It ranged from –0.9 to 2 ppb at AMY and from –1.25 to 0.84 ppb at JGS. ULD had a two week calibration periods, which changed to one month from May 18, 2017 to November 11, 2019. During this period of one month calibration frequency, $U_r$ increased to a

maximum of 4 ppb, which is greater than the WMO/GAW compatibility goal of ±2 ppb. After conducting the reproducibility test in November 2019, the calibration frequency decreased to 5 days. Therefore, $U_r$ at ULD was separated into two groups including or excluding the period with a longer calibration period (with asterisk in Table 3). When we considered only the period with a higher calibration frequency, the uncertainty at ULD was similar to that at other stations. This means that $U_r$ is the largest component of measurement uncertainty, and that $U_T$ can be decreased using an appropriate calibration strategy.

$U_p$ was determined from the standard deviations of working standard measurements, as described in section 2.3, and expressed by a pooled standard deviation (Eq. 5).

$$U_p = \sqrt{\frac{\sum_{i=1}^{N} Ni \times Si^2}{Ni - Nt}}$$ (Eq. 5)

where $S_i$ is the standard deviation of 10 min averages of working standard measurements; $N_i$ the index number of a measurement during 10 min (based on 5 s intervals); and $N_t$ is the total number of calibrations during the period. $S_i$ was less than 0.882 ppb at AMY, 0.603 ppb at JGS, and 0.688 ppb at ULD. The pooled standard deviations ($U_p$) are shown in table 3.

According to Zhao et al. (2006), the uncertainty of working standards can be calculated by the propagation error arising from the uncertainty of primaries with a maximum propagation coefficient ($\gamma = 1$) and repeatability. Similarly, $U_{scale}$ for working standards is determined by (Eq. 6)

$$U_{scale} = \sqrt{U_p^2 + U_{lab}^2}$$ (Eq. 6)


where $U_{lab}$ is the uncertainty of laboratory standards, which CCL (NOAA/GML) certified. Here, $U_{lab}$ has the same value as the uncertainty of secondary standards, 0.3 ppb with a confidence interval of 68%, based on calibration of the secondary standards against the primary standards (http://gml.noaa.gov/ccl/ch4_scale.html, last access: January 2022). These values were the same for all stations since they were calibrated by a central lab at NIMS in Jeju. Therefore, $U_p$ is the repeatability at the central lab since we

propagated the standard scale through the same analyzer and set-up for atmospheric monitoring. This value was always less than 0.12 ppb.



Table 3. Uncertainty estimates for measurements of $CH_4$ at each station from 2016 to 2020. Units are ppb. All terms are 68% confidence intervals

| Uncertainty terms | AMY | JGS | ULD |
|---|---|---|---|
| $U_{h2o}$ | 0.006 | 0.006 | 0.008 |
| $U_p$ | 0.157 | 0.120 | 0.351 |
| $U_r$ | 0.578 | 0.365 | 2.323 (0.710*) |
| $U_{scale}$ | 0.323 | 0.323 | 0.323 |
| $U_T$ | 0.778 | 0.728 | 2.352 (0.801*) |

*This value was calculated excluding the period with one-month calibration frequency

For AMY, the difference from the CCL in the RR test with the analysis of the same cylinder was from –0.3 to 1.3 ppb with GC-FID in 2010. Therefore, we considered the largest value of 1.3 ppb as the measurement uncertainty from 1999 to February 2016 during the GC-FID measurement period.

Overall, the total measurement uncertainty was calculated to be from 0.728 to 0.801 ppb. These values were similar to those

reported by CRDS measurements (< 1 ppb) (Winderlich et al., 2010; Andrews et al., 2014, Verhulst., 2017). In the future, quoted uncertainties could be greater owing to the inclusion of more error sources, while repeatability and reproducibility may improve with a different calibration strategy.

### 3.2 Local/regional effects on observed $CH_4$

The enhancement of $CH_4$ relative to the regional background can help evaluate local/regional additions to $CH_4$, with the excess signal defined as (Eq. 7):

$$CH_{4XS} = CH_{4OBS} - CH_{4BG} \qquad \text{(Eq. 7)}$$

where $CH_{4OBS}$ is L2 hourly data (before filtering) and $CH_{4BG}$ indicates the regional background at a site as determined by the smoothed curve fitted to L3 daily data (section 2.4.2). $CH_{4XS}$ was greatest in the order AMY (55.3±37.7 ppb) > JGS (24.1±10.2 ppb) > ULD (7.4±3.9 ppb) from 2016 to 2020. For ULD, we excluded data collected in 2016 as they were affected by the garbage incinerator next to the station (section 2.4, Figure 2(c)). All stations showed largest $CH_{4XS}$ in summer (June, July, August) with 109.6 ± 23.8 ppb at AMY, 37.0 ± 2.1 ppb at JGS and 12.2 ± 3.7 ppb at ULD. Conversely, the smallest values were observed as

25.6 ± 2.4 ppb at AMY and 18.8 ± 4.1 ppb at JGS in spring (March, April, May), while the lowest value of 7.5 ± 0.4 ppb at ULD was observed in winter (December, January, February). The baseline selection conditions listed in Table 2 also supported this result. The selected baseline data accounted for only 11–37.6% of summer data at all stations, indicating that $CH_4$ levels were





elevated in summer. In winter and spring, we could better capture well mixed air compared with other seasons (28.4–58.9%) because of the strong westerly wind with Siberian high.

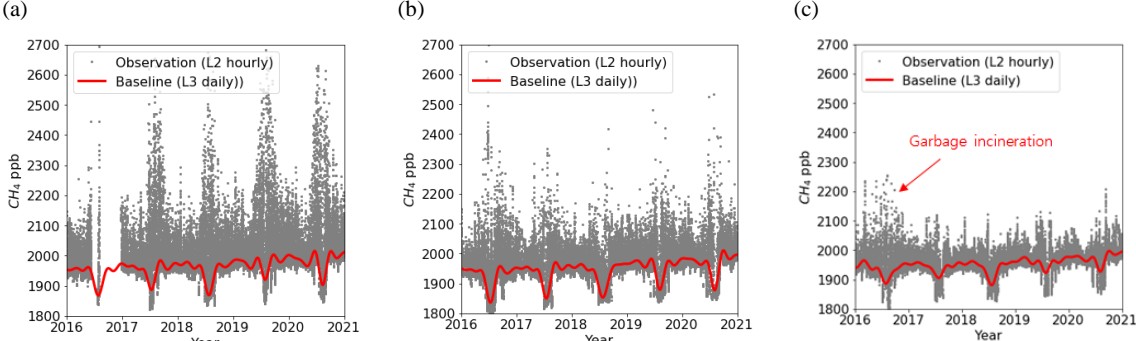

Figure 2. L2 hourly (grey scatters, observation) and fitted L3 daily data (red line, Baseline) at (a) AMY, (b) JGS, and (c) ULD from 2016 to 2020.

To understand the influence of local surface wind on observed $CH_4$, bivariate polar plots were used for 2018. These plots express the dependence of all hourly $CH_4$ (L2 hourly data) on wind direction and speed (Fig. 3–5). The wind data were derived from an

AWS, as described in section 2.1.

At AMY, when wind speed was consistently $< 3$ m·s$^{-1}$, $CH_4$ was elevated during all seasons. Especially in summer, it showed strong signals when wind direction was between 45° and 135° (from land). A similar observation was made in other seasons, possibly indicating that this was related to local influences such as from rice paddies. The dominant wind direction was south-west in summer and north-west in winter. Even though lower $CH_4$ levels were captured regardless of wind direction with increased wind

speed, $CH_4$xs was still higher than that at the other two stations. Therefore, AMY could be affected not only by local activities but also by distant emissions.

JGS experienced the strongest winds among the three stations in all seasons (maximum 27.5 m·s$^{-1}$). Strong north-westerly wind (open sea) occurred in spring and winter, and air masses from the north-east (Korean inland) were noted during autumn and from the south (open sea) during summer. $CH_4$ was lower than at AMY, with strong signals observed in all seasons under different

criteria. Higher $CH_4$ levels occurred because of winds from the eastern part of JGS in autumn and summer when the wind speed decreased to less than 5 m·s$^{-1}$. For spring and winter, strong signals were noted in the eastern and northern parts of JGS. Since JGS is located downwind from continental Asia and strong westerly winds occur in winter/spring because of the strong Siberian high, this signal might be related to activities not only in Asia but also in Jeju.

For ULD, the main wind directions were quite clearly from 0° to 90° (30%) and from 180° to 270° (33%), and wind speeds less

than 5 m·s$^{-1}$ occurred 72% of the total time. High $CH_4$ episodes were mainly observed when the wind direction was between 180° and 225°, presumably affected by the southeastern part of the Korean Peninsula. This wind direction was very dominant in summer with a lower wind speed than that in other seasons.

Overall, atmospheric $CH_4$ observed by KMA GAW stations was affected not only by the local area but also by air masses from continental Asia, as indicated by the results from synoptic systems. Signals at AMY may be affected by local/regional activities,





such as agriculture and livestock industries, owing to the relatively lower wind speeds; however, it still showed higher values compared with those of other stations when it captured well-mixed air. This indicates that AMY was affected not only by local sources but also by long range transport of air masses originating from continental Asia. ULD showed lower $CH_4$ and was less affected by local impacts.

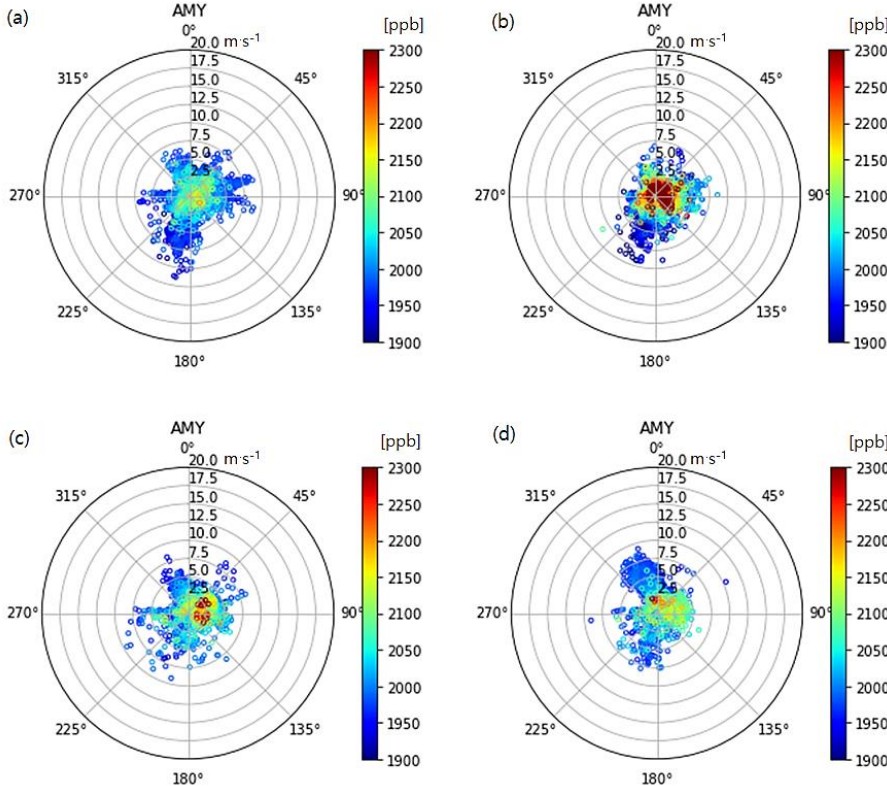

Figure 3. Bivariate polar plots for observed $CH_4$ (L2 hourly) in spring (a), summer (b), autumn (c), and winter (d) at AMY in 2018.





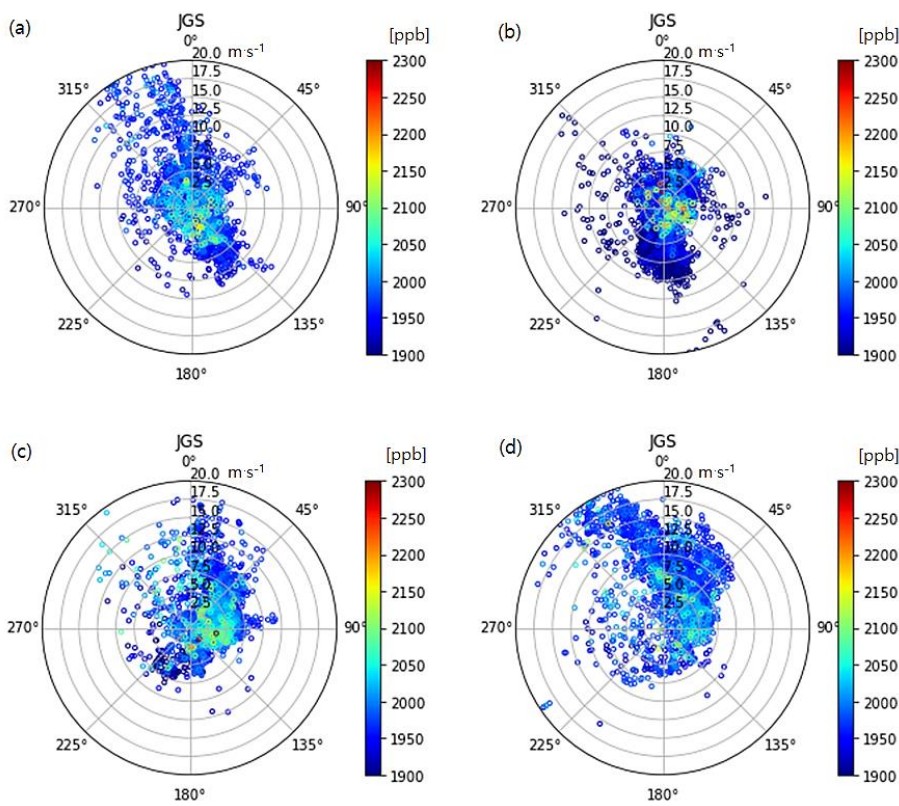

Figure 4. Bivariate polar plots for observed CH$_4$ (L2 hourly) in spring (a), summer (b), autumn (c), and winter (d) at JGS in 2018.





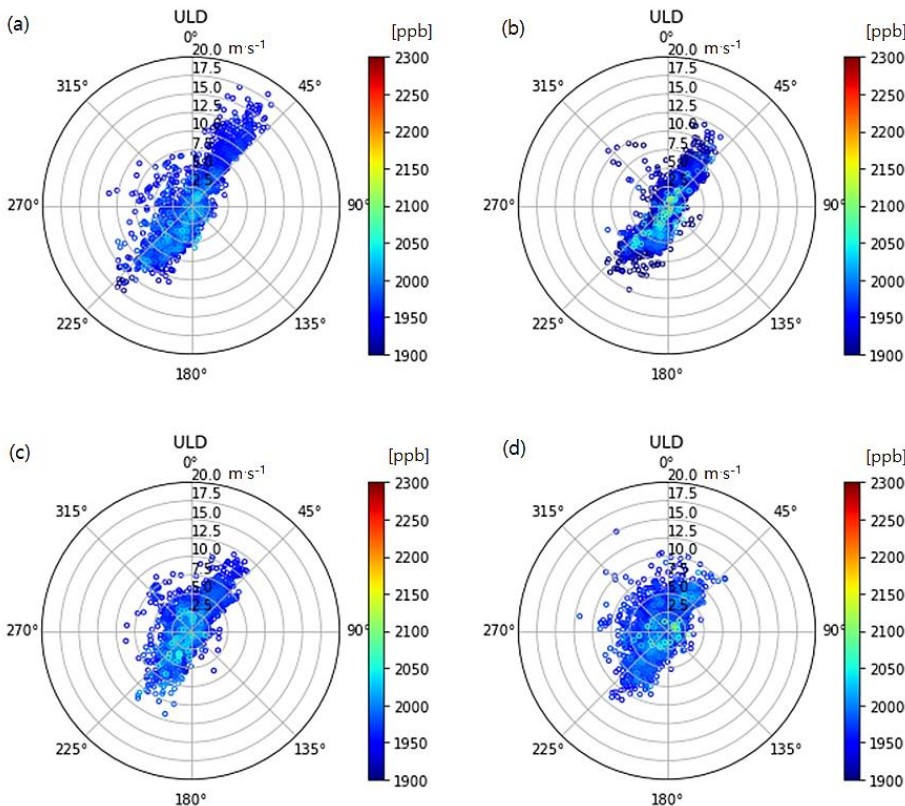

Figure 5. Bivariate polar plots for observed $CH_4$ (L2 hourly) in spring (a), summer (b), autumn (c), and winter (d) at ULD in 2018.

### 3.3    Average diurnal variation

Diurnal $CH_4$ variations were calculated as the average deviation from the daily mean in each month from L2 hourly data from 2016 to 2020 (recent 5 years) for AMY and JGS, and from 2017 to 2020 for ULD (Fig. 6).

Among the three stations, the mean diurnal variation of all seasons was greatest at AMY (69.5±49 ppb) and smallest at ULD

(7.6±4.2 ppb), while it was 24.7±14.4 ppb at JGS. Daily variations in $CH_4$ are generally small at global scale stations, so stations

with a large seasonal cycle amplitude may be affected by local/regional sources (Aoki et al., 1992) and transport driven such as

upslope/downslope air and land/see breeze due to geographical reason.

AMY was surrounded by $CH_4$ sources as described in the introduction, while ULD had similar characteristics to global scale stations that are less impacted by their local/regional environment. Similar to ULD, Mt. Waliguan station (3816 m), a representative global GAW station in Asia, also showed an amplitude of 5 to 10 ppb for the diurnal $CH_4$ cycle (Zhou et al. 2004., Fang et al., 2013).

Atmospheric $CH_4$ at AMY and JGS started to increase around midnight and peaked from 5 to 8 AM local time and then decreased with minimum value from 15 PM to 17 PM. For ULD, the peaks were observed between 6 to 11 AM, especially in summer, but there were no significant troughs. These variations at AMY and JGS were consistent with the changes in wind pattern and BLH.





BLH was maximum near the middle of the day. At night, radiation loss at ground level leads to a stable boundary layer, leading to accumulation of $CH_4$ (Worthy et al., 1998, Higuchi et al., 2003). Both stations were also affected by land-see-breeze and received air from seaside during the daytime, which enhanced the diurnal variation. These patterns of $CH_4$ were similar to those of $CO_2$ observed by both stations (Lee et al., 2019) because of similar meteorological conditions. However, ULD is located on the slope

5    of the mountain and is surrounded by a complex terrain, thus being affected by certain winds from north to east and south to west, regardless of time and season. However, peak values only occurred during 6 to 11 AM, which needs further study.

All stations showed the lowest amplitude in winter (Dec. to Feb.) and the largest amplitude in summer (Jun to Aug.). AMY showed the largest diurnal amplitude (176.8±74.6 ppb) in August among the three stations, which was almost 2.5 times greater than the

annual mean value, with substantial variation among months. JGS and ULD showed the largest amplitude in August (43.4±17.7

10    ppb) and July (17.2±11.6 ppb), respectively. This indicated that the emission and meteorological impacts (e.g. maximized BLH

and land-see breeze) were strong in summer. AMY is close to rice paddies (110 km$^2$), which are the major source of $CH_4$ in summer. JGS and ULD are not close to waterlogged paddies. However, high temperatures stimulate greater emissions from sources such as agriculture, livestock, and wetlands, thus affecting emissions at both stations. Similar to the observation made at AMY, previous studies have shown large variation in $CH_4$ emissions from the rice paddy area (196±65 ppb) and wetland (~150 ppb) during

15    summer (Worthy et al. 1998., Fang et al., 2013).





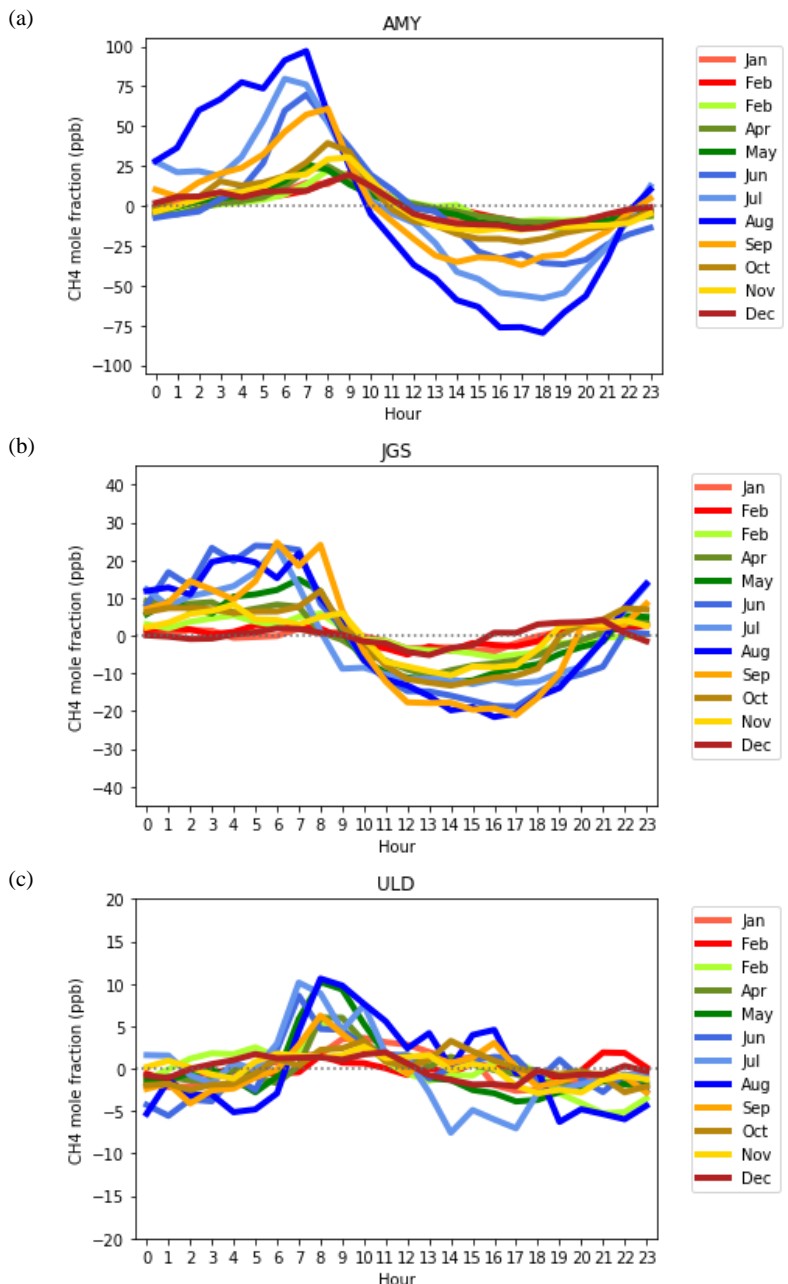

Figure 6. Mean diurnal variations of CH₄. Values show the average departure from the daily mean in each month at (a) AMY and (b) JGS from 2016 to 2020 and (c) at ULD from 2017 to 2020.





### 3.4 Comparison with other East Asian stations: annual mean, seasonal amplitude, and growth rate

#### 3.4.1 Annual mean

Time series of monthly mean $CH_4$ from KMA's three stations and the two other stations in East Asia are compared in Fig. 7(a) and annual mean $CH_4$ is summarized in Table 4. Annual $CH_4$ mean was the highest at AMY and the lowest at WLG. The other stations showed similar levels of annual mean considering standard deviations. It was obvious that AMY was affected by local and regional activities, while WLG was affected by representative air masses of the Northern Hemisphere.

#### 3.4.2 Seasonal amplitude

As described in section 2.4, the seasonal amplitudes of $CH_4$ from 2016-2020 was calculated for three KMA stations and are compared with those values from WLG and RYO (Table 4). The seasonal amplitude is related to the combination of $CH_4$ surface flux distribution and chemical loss by reactions with OH and by soil loss. Seasonal amplitudes followed the order JGS > AMY > ULD>RYO> WLG (Table 4).

Since WLG is a global baseline station that is affected less by regional sources and sinks, the amplitude was smaller compared with the other regional stations. The amplitude of WLG was similar to that of other global stations such as Mauna Loa (30.6±4.2 ppb) (Dlugokencky et al., 1995). Seasonal amplitudes at AMY and JGS are much greater than at other 3 stations, and even inland regional stations in China, such as Lin'an (77±35 ppb) and Longfenshan (73±8 ppb) (Fang et al., 2013). Minimum values at JGS are -14.8±9.2 ppb lower than AMY minimum values while the maximum values at both stations are similar. Li et al. (2018) reported the summer airmass was affected by large-scale low-level monsoonal circulation across the tropic in Jeju. A similar meteorological impact for $CH_4$ was reported at TAP (28 km away from AMY) (Dlugokencky et al., 1993). Even though transport and OH radical can result in low $CH_4$ values at AMY during, the station is also affected by nearby sources with enhanced emissions during summer. As we introduced in section 2.1, AMY has large rice paddies and livestock industries within 50 km. During summer, high temperatures will enhance $CH_4$ emissions from these sources, leading to higher $CH_4$ than that at JGS (Kenea et al., 2021; Wang et al., 2021). Among regional stations, ULD and RYO may have been less affected by regional flux because of their altitude, causing their amplitude to be greater than that of WLG but smaller than that of AMY or JGS.

Minimum values were observed in summer, while maximum values occurred in spring, autumn, or winter for different regional stations. In contrast, WLG showed maximum levels in summer and minimum values in winter/spring. Zhang et al. (2013) reported that regional/local sources and air masses from polluted regions influenced by industry, crop residue burning, and agriculture may affect $CH_4$ observations at WLG in summer.





(a)     (b)

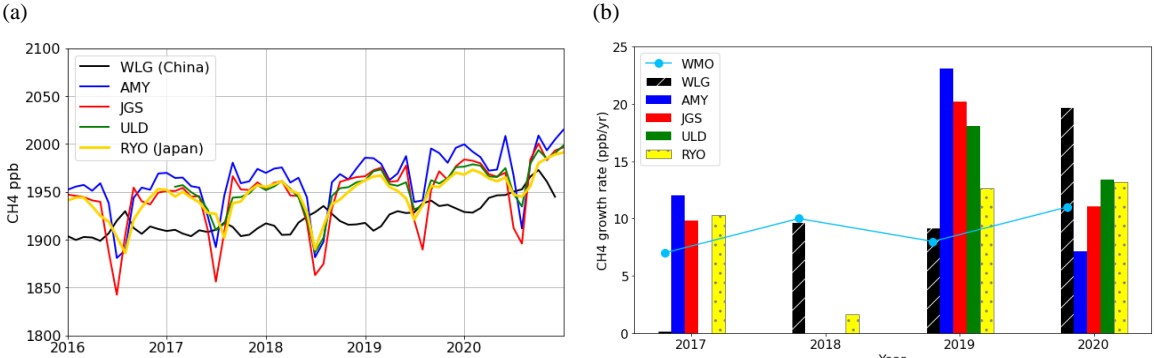

Figure 7. Time series of (a) monthly mean CH$_4$ and (b) annual growth rate at WLG, AMY, JGS, ULD, and RYO. The growth rate reported by WMO (WMO, 2021) is overlaid on (b) and this value is calculated as the change in annual mean from the previous year.

Table 4. Annual mean CH$_4$ with standard deviations from monthly mean from 2016 to 2020, mean seasonal amplitudes, and growth rates. Seasonal cycle amplitude during each calendar year was magnitude of the peak to trough of the detrended seasonal cycle (see section 2.4.2). The growth rate is an annual increase (not de-seasonal), absolute difference from previous year. Growth rate at ULD was calculated only from 2017 to 2020. Units are dry-air mole fractions (ppb)

| Year | WLG | AMY | JGS | ULD | RYO |
|---|---|---|---|---|---|
| 2016 | **1909±9** | 1942±28 | 1928±33 | - | 1929±19 |
| 2017 | **1909±4** | **1954±24** | **1938±31** | **1942±15** | **1939±15** |
| 2018 | 1919±9 | **1953±31** | **1937±35** | 1941±24 | **1941±21** |
| 2019 | 1928±10 | 1976±19 | 1957±26 | 1959±13 | 1954±14 |
| 2020 | 1948±14 | 1983±27 | 1968±32 | 1972±17 | 1967±14 |
| Mean seasonal amplitude over 5 years. | 21±5 | 100±13 | 118±9 | 67±12 | 58±8 |
| Mean annual growth rate over 5 years (ppb·yr$^{-1}$) | 10±8 | 10±10 | 10±9 | 10±10 | 10±5 |

### 3.4.3 Growth rate

The annual growth rate was calculated by absolute difference from previous year (not de-seasonal) (Fig.7 (b)). When we analyzed the overall growth rate for 5 stations, mean values of annual growth rate from 2017 to 2020 were around 10 ppb·yr$^{-1}$, which was similar to the WMO global mean (9±2 ppb·yr$^{-1}$). However, when yearly comparisons were made, WLG and the other 4 regional

15    stations varied. Especially from 2016 to 2017, WLG showed no increase in CH$_4$, while CH$_4$ at other stations increased from 2017





to 2018. Normally the growth rate in CH₄ at WLG matches well with the WMO global trend (Fig. 7. (b)). Wang et al.(2021) reported that CH₄ fluxes in Asia are influenced by ENSO and temperature; therefore, we compared the growth rate of CH₄ from four regional stations with both factors (Fig. S2). This showed that the pattern of growth rate was quite similar to that of ENSO; however, even though the ENSO was negative (e.g., from 2017 to 2018) when surface temperature was high, the growth rate still

increased. Plots of $\delta^{13}C_{(CH4)}$ vs 1/CH₄ had intercepts (indicating $\delta^{13}C_{(CH4)}$ source signatures of the enhancements) of –52.3±2.2‰ in winter and –53.7±0.7‰ in summer at AMY during 2016 to 2020 (Fig. S3). These values are very similar to the observed values during the summer vegetation period when biogenic emissions are very active (–52.5±1.9‰) in Europe (Varga et al., 2021), indicating that AMY was mainly affected by biogenic sources regardless of the season during this period. Throughout Asia, emissions from agriculture and waste account for over 50% of the total, which is increasing every year (Jackson et al., 2020). Since

climate variability such as ENSO and temperature drive biogenic sources, the CH₄ growth rates observed at regional stations in Asia are more sensitive to regional emissions than the global station.

### 3.5    Long-term records of CH₄ and its drivers in East Asia

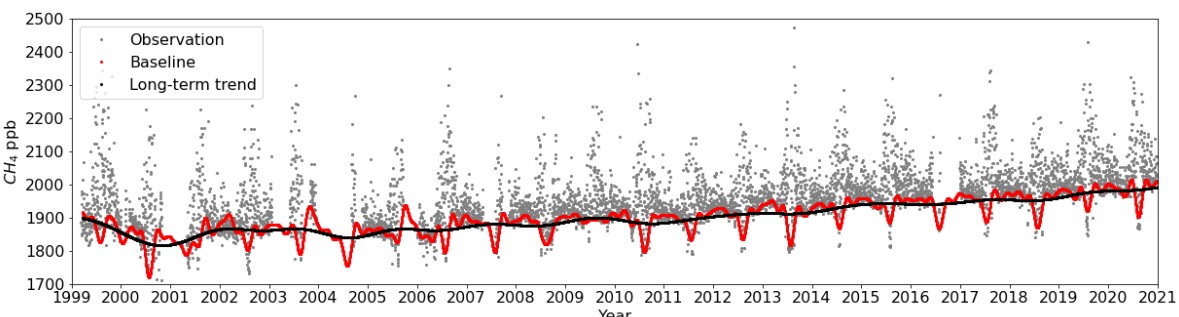

Figure 8. Time series of CH₄ (L2 daily, grey), baseline (L3 daily, red), and long-term trend (black) observed at AMY from 1999 to 2020.

The CH₄ trend is defined as upward growth in the data with the seasonal cycle removed (Thoning et al., 1989) . The long-term

trend at AMY was very similar to the global trend. From 1999 to 2005, the mean annual CH₄ growth rate (absolute differences from the previous year) was approximately –1.2 ppb·yr⁻¹ at AMY, while the global value was 0.3 ppb·yr⁻¹ and both values have increased since 2006. CH₄ increased by 3.3 ppb·yr⁻¹ from 2006 to 2010 (global: 5.9 ppb·yr⁻¹) and by 8.3 ppb·yr⁻¹ (global : 9 ppb·yr⁻¹) from 2016 to 2020, indicating that the growth rate is accelerating. CH₄xs did not vary much and was 49±74 ppb during 2006–2010 and 50±70 ppb during 2016–2020.

For PSS analysis with hourly CH₄xs, however, the source distribution changed between 2006–2010 and 2016–2020 (Figure 9 (a) and (b)). We found that CH₄xs from CS and KL were enhanced and remain constant respectively while that from CN became weaker in 2016–2020 compared with that in 2006–2010. This means that source region characteristics related to CH₄xs had



changed. Sources affecting CS and KL paddy fields and that for CN was reported to be fossil fuel emissions (Zhang et al., 2011, Ito et al., 2022).

Using TAP $\delta^{13}C_{(CH4)}$ long-term measurements, keeling plots indicated that emissions from CN were mainly related to fossil fuel or biomass burning (–40.5±3.2‰), while CS (–56.9±3.2‰) and KL (– 54.6±2.3‰) were affected more by biogenic sources

during 2006–2010 (Fig.9 (c) and (d)). Sherwood et al. (2017) reported unweighted global mean $\delta^{13}C$ of –44.8±10.7‰ from fossil fuel use, –26.2±4‰ from biomass burning, and –61.7±6.2‰ from microbial sources. Even though the uncertainty of isotopic source signature is quite large, $CH_4$ formed at high temperature such as combustion is enriched in the heavier isotope while $CH_4$ from wetland, rice paddies and livestock is depleted. Air masses from CN were affected by several sources, however, this area was mainly related to pyrogenic and fossil fuel emissions as per the enriched $\delta^{13}C_{(CH4)}$ compared to that of other sectors.

During 2006–2010, $\delta^{13}C_{(CH4)}$ at CN and KL presented no significant trend, while the value observed at CS decreased.

However, we can confirm that $\delta^{13}C_{(CH4)}$ in all three sectors showed a range from –52.2±2.7‰ to –49.5±2.7‰ during 2016-2020, indicating that the $CH_4$ sources in East Asia were mostly biogenic sources, although CN presented a fossil-fuel related source. The decreasing trend of $\delta^{13}C_{(CH4)}$ at CN, CS, and KL also supports this assumption with lower values compared with those in 2006–2010 (Figure 9.(e) and (f)). Since coal emissions decreased from 2010 in China (Liu et al., 2021) and temperature impacts

biogenic $CH_4$ emissions, the major drivers of $CH_4$ may have shifted from fossil fuel/biomass burning to biogenic sources such as wetlands and rice paddies.

Overall, AMY and global growth rates were renewed in 2006 and during 2006–2010; the increasing trend could be linked to mixed biogenic and fossil fuel sources in East Asia. However, the recent accelerated increase in $CH_4$ emissions during 2016–2020 is more related to biogenic sources such as agriculture and wetland (Jackson et al., 2020, Lan et al., 2021).





Figure 9. PSS analysis from (a) 2006 to 2010 and (b) 2016 to 2020. "Keeling" plots for (c) 2006 to 2010 and (d) 2016 to 2020. Time series of $\delta^{13}C_{(CH4)}$ from (e) 2006 to 2010 and (f) 2016 to 2020.



## 4.    Summary and Conclusions

Among greenhouse gases, $CH_4$ emission reductions can be highly effective for short-term global warming mitigation because of its relatively short lifetime. However, its sources are diverse and yet to be evaluated completely through high accuracy measurements. Our study analyzed $CH_4$ characteristics observed at regional KMA GAW stations, uncertainties related to its measurement, and changes in sources using long-term data in South Korea.

KMA started monitoring atmospheric $CH_4$ in 1999 at AMY and expanded its network to the south and east parts of Korea at JGS and ULD in 2012 using a new system consisting of CRDS and a cryogenic drying system.

All three stations have similar measurement uncertainty from 2016 to 2020, in the range of 0.728–0.801 ppb. These uncertainties are similar to values reported in previous studies (less than 1 ppb). In addition, we confirmed reproducibility as the greatest contributor to measurement uncertainty; therefore, calibration strategies are the most critical component for reducing measurement uncertainty.

$CH_4xs$ assessed relative to local background levels at each station was in the order AMY ($55.3\pm37.7$ ppb) > JGS ($24.1\pm10.2$ ppb) > ULD ($7.4\pm3.9$ ppb). $CH_4xs$ was greatest in summer and lowest in spring or winter. This result is consistent with wind direction and speed. In summer, local biogenic sources affected observed $CH_4$. Low wind speed enhanced $CH_4xs$, while lower $CH_4$ levels were observed when the stations experienced high wind speed. For AMY, even when $CH_4$ was measured in well-mixed air, its level was higher than that at other stations, indicating that it was affected not only by local sources but also by distant air masses from Asia. ULD showed representative $CH_4$ levels without local impacts. Diurnal variations were greatest at AMY and smallest at ULD, and were affected by local sources and meteorological characteristics. The variation at ULD was $7.6\pm4.2$ ppb, a value similar to that at the WLG baseline station in China (5 to 10 ppb). All stations had large diurnal cycles in summer, indicating a strong influence of local biogenic sources.

When $CH_4$ seasonal cycle amplitudes measured at KMA stations were compared with those at other East Asian stations from 2016 to 2020, the following descending order was observed: JGS ($103\pm10$) > AMY ($85\pm16$) > ULD ($58\pm12$) and RYO ($57\pm12$) > WLG ($30\pm11$). As discussed, since AMY reflected strong local influences not only in winter but also in summer, its seasonal amplitude was smaller than that of JGS. However, the annual $CH_4$ mean was the highest at AMY and lowest at WLG. The relative contributions of $CH_4$ source types to signals at regional stations in Asia are sensitive to temperature and ENSO. Based on analysis of $\delta^{13}C_{(CH_4)}$ measurements, we established an increase in $CH_4$ from biogenic sources.

From the long-term analysis of $CH_4$ data at AMY, average $CH_4$ growth rate was 3.3 ppb·yr$^{-1}$ during 2006–2010, but increased to 8.3 ppb·yr$^{-1}$ in 2016–2020. However, $CH_4xs$ was similar while the source distribution was different based on our PSS analysis during each period. We infer significant changes in air masses arriving from our northern China sector. During 2006-2010, the main sources contributing to the $CH_4$ observations were fossil fuel and pyrogenic. The main sources shifted to biogenic/natural during 2016–2020. $CH_4$ emissions in the southern part of China and in local regions in Korea also had an enhanced biogenic/natural source signal.





*Data availability*
Atmospheric CH$_4$ data in KMA network are from HL and can be downloaded from climate.go.kr/home/09_monitoring/search/search. TAP CH$_4$ and $\delta^{13}$C$_{(CH4)}$ data can be downloaded from https://doi.org/10.15138/VNCZ-M766/. RYO (DOI, https://gaw.kishou.go.jp/search/file/0001-2012-1002-01-01-9999) and WLG
(DOI, https://doi.org/10.15138/VNCZ-M766) data are downloaded from World Data Centre for Greenhouse Gases.

*Author contributions*
HL designed, wrote this paper and analyzed the data. WIS implemented the data QA/QC. SL implemented PSS analysis. SL run the central calibration centre for KMA stations and provide the calibration strategy. WIS, SL, SL, SK and SJ reviewed the
manuscript. All authors contributed this work.

*Acknowledgments.*
We appreciate all staff and technicians at AMY, JGS, ULD in the Korea meteorological network, WLG in China and RYO in Japan. Also special thanks to Dr. Ed Dlugokencky at NOAA and Dr. Silvia Michel at Colorado University for the support of this
study. This work was funded by the Korea Meteorological Administration Research and Development Program "Development of Integrated Climate Change Monitoring and Analysis Techniques" under Grant (KMA2018-00324).

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
