# Peer review of "Measurement report: Atmospheric CH4 at regional stations of the Korea Meteorological Administration/ Global Atmosphere Watch Programme: measurement, characteristics, and long-term changes of its drivers"

_Atmospheric Chemistry and Physics, 2022_

## Referee Comment (RC1)

**Review's comments**

**Manuscript Number:** *acp-2022-600*

**Title:** Measurement report: atmospheric $CH_4$ at regional stations of the Korea Meteorological Administration/ Global Atmosphere Watch Programme: measurement, characteristics and long-term changes of its drivers

**Authors:** Lee H., Seo W.-I., Li S., Lee S, Kenea S, and Joo S.

The authors of this study present high-quality record of the atmospheric $CH_4$ mole fractions measured at three WMO/GAW stations in Korea: Anmyeondo (AMY), Jeju Gosan Suwolbong (JGS), and Ulleungdo (ULD). To confirm the data quality in detail, the authors evaluate the total measurement uncertainties at the three sites, which are less than GAW's compatibility goal for atmospheric $CH_4$ measurement ($\pm 2$ ppb). Then, they investigate the characteristic features of the $CH_4$ variations by precisely analyzing the excess $CH_4$ mole fractions from the background variations, the diurnal cycles, the seasonal cycles, and the growth rates. From PSS analysis and the $^{13}CH_4$ isotopic data obtained at nearby site (TAP), they also discuss the temporal changes in the source region and the dominant source categories from a 5-year period of 2000-2006 to that of 2016-2020. Although I think the discussion based on the PSS and isotopic analyses have some problems, I believe that the data presented in this report are quite reliable and contribute to the global modeling studies especially for the $CH_4$ emission analysis from the east Asia.

I found that the paper contains material that should be published in Atmospheric Physics and Chemistry. However, I think that the authors should make more efforts to improve the manuscript as is suggested in followings before publication.

**General comments:**

1) I'm not sure why the authors discuss in depth the influence of water vapor on the CRDS analyzer in Section 2.3 and 3.1. The CRDS analyzer provides dry $CH_4$ mole fraction by measuring the water vapor simultaneously. The study of Rella et al. (2013) revealed that the preliminary reduction of water vapor of sample air less than

1% allow CRDS analyzer to measure the $CH_4$ mole fraction within the level of GAW's compatibility goal ($\pm 2$ ppb). The authors describe that the air samples are dried to $-50°C$. I believe that the laboratory standard gases and working standard gases uses in this study are sufficiently dried (I think that the absolute dew points should be clarified in the text). In addition, if the air samples are dried by a cold trap of $-50°C$, the dilution effect is comparable to the reproducibility of the CRDS analyzer. Therefore, I think it would be better to simplify the description about $H_2O$ influence on the CRDS measurement.

2) I'm very curious about the diurnal cycles of the L3 hourly data. If the data selection method described in Section 2.4.2 effectively remove the local effect, there is no statistically significant diurnal cycles in the L3 data. Such information would be useful to make the L3 data more reliable.

3) I don't agree with the discussion based on the PSS and the $\delta^{13}CH_4$ analyses in Section 3.5.

As for the PSS analysis, I'm not sure how to interpret the plots shown in Figs 9a and 9b. Although the authors seem to consider that the plots reflect the $CH_4$ flux distributions, the flux distribution is considerably different from the previously reported flux distributions (e.g., Ito et al., 2019). Additionally, I cannot believe that such large temporal change in the flux distribution occurred between the two periods. So, I think that the authors should clarify what the plots based on the PSS analysis mean. The authors only show the plots based on the observation at AMY. But I'm curious about the similar plots based on the data at JGS and ULD. If the distributions based on JGS and ULD are similar to Fig. 9b, such result would convince us to some extent that the PSS analysis is reliable.

I'm not sure what the isotopic analysis means. The authors would like to suggest that influences of the biogenic sources increased from 2006-2010 to 2016-2020 because the $\delta^{13}C_{(CH4)}$ values for the latter period shows faster decreasing rates than those for the former period. However, the background $\delta^{13}C_{(CH4)}$ data also show the secularly decreasing trend (for example, you can see the recent $\delta^{13}C_{(CH4)}$ change in NIWA's home page: https://niwa.co.nz/atmosphere/our-data/trace-gas-plots/methane). So, how do the

authors distinguish the regional influence on the $\delta^{13}C_{(CH4)}$ values from the background change? Additonally, why don't the authors use Miller-Tans plot to evaluate of the $\delta^{13}C$ values for the $CH_4$ regional sources? On the other hand, the values of the intercept of the Keeling plot for CS and KL increased from 2006-2010 to 2016-2020, suggesting the influence of the biogenic sources relatively decreased. This result is inconsistent with the above result.

I'm also curious about the $\delta^{13}C_{(CH4)}$ data during the period from 2010 to 2016. Why don't the authors discuss the period from 2010 to 2016?

**Specific comments:**

Page 2, line 28: "Kim et al., 2014" should be "Kim et al., 2015".

Page 2, line 30-32: "Li et al., 2020" and "Li et al., 2022" are not listed in References.

Page 2, line 36: "Turnbull et al., 2015" is not listed in References.

Page 3, line3: "Watanabe et al. 2000" is not listed in References.

Page 4, line 17: What's the "−90℃" mean? If it's not the trap temperature, it should be removed to avoid confusion.

Page 5, line 6: I believe that the four laboratory standards are kept in high-pressure cylinders. I think that it would be better to clarify what kind of cylinders are used for the laboratory standards (volume and material).

Page 5, line 13-20: I'm a little bit confused by this part. I think that the CRDS analyzer (Picarro) give dry mol fractions of the atmospheric $CH_4$. So, why does the authors discuss the dilution effect of water vapor?

Page 5, line 17 (Eq. 1): It should be better to clarify what the $C_{dilution}$ and $C_{dry}$ mean. And the definition of $H_{act}$, "$H_2O$ difference between standard gases and samples", is rather confusing. Is it "$H_2O$ difference between laboratory standard gases and working standard gases"?

Page 7, line 10-11: It should be better to clarify the detail of the fitting function (order of polynomial and order of harmonics) and the cut-off frequency for the digital filtering based on Thoning et al. (1989).

Page 7, line 12: It should be better to clarify how to compute the monthly data, long-term trend, and seasonal amplitude in detail.

Page 7, Table 2: There are no values of MS for AMY and ULD. Does it mean that MS criteria were not applied to the data for AMY and ULD?

Page 7, line 25-26: The geological locations of WLG and RYO are already given in page 3, line 1-3.

Page 8, line 6-8: It should be better to add the reference for the $CH_4$ isotopic measurement at INSTAR.

Page 8, line 5 and 9: It says here that the distance between TAP and AMY is 24 km. But other parts in this manuscript, the distance is 28 km. Which value is correct?

Page 8, line 9-10: It should be better to add a figure showing synoptic scale variations of $CH_4$ observed at AMY and TAP.

Page 8, line 35-page 9, line 4 (2.7 PSS analysis): Since I'm not sure PSS analysis, I tried to read the cited papers, Remann et al., 2004, 2008; Li et al., 2017. But these papers are not listed in References. If the single line for each backward trajectory is used for the calculation of $C_{(i,j)}$, the sensitivity of the potential source strength increase with the distance from the start point in comparison with the reality. This effect would result in the overestimation of the potential source strength with increasing the distance from the station, wouldn't it?

Page 12, line 8-9: Are the characteristics of the bivariate polar plots for the three stations (Figs. 3-5) same with those for the other years?

Page 18, line 10-11: The seasonal cycle of the atmospheric $CH_4$ is also influenced by the atmospheric transport.

Page 18, line 19: The distance between TAP and AMY is repeatedly described in this manuscript.

Page 18, line 20: "during" what?

Page 19, Figure 7b: Why don't the authors plot the negative growth rate in the figure?

Page 19, line 12: Was the annual growth rate calculated from the annual means based on the monthly means, listed in Table 4? If so, it would be helpful for the readers to describe that in the manuscript.

Page 19, line 15: $CH_4$ at AMY, JGS, JLD, and RYO showed significant increases from 2016 to 2017, but small (or negative) increases from 2017 to 2018.

Page 20, line 1: There is relatively large differences in the $CH_4$ growth rate between WLG and the WMO global mean in 2017 and 2020.

Page 20, line 5-6: There is no description about the $\delta^{13}CH_4$ data at AMY. Please add the explanations for the data. In addition, since the plot shown in Fig. S3 is not Keeling plot but Miller-Tans plot, the slopes of the plots represent the $\delta^{13}C$ of the $CH_4$ sources.

Page 21, line 1: "Sources affecting CS and KL were paddy fields and …" ?

Page 21, line 13: I'm not sure what "this assumption" means.

Page 21 line 3-14: Are all the flask data collected at TAP plotted in Figs 9? Are the source regions of all the flask data classified into three sectors, CS, CN, and KL? How are they classified?

Page 26, line 21-23: "Shuang-Xi et al., 2013" is not cited in the text.

Page 27, line 3-5: "Winderlich et al., 2010" is not cited in the text.

---

## Referee Comment (RC2)

**Review of Lee et al., ACPD https://doi.org/10.5194/acp-2022-600**

The clear explanation of the set up of the measurements at the 3 stations, scale propagation and uncertainty would be valuable to others developing and refining greenhouse gas measurement systems. The publication of the high quality data from the stations, and discussion of the regional context is good. I think further work should be done on the source identification (section 3.5). This may change the overall conclusion of the work – i.e. whether the changes in the region are a relative increase in biogenic sources or not.

Before publication some edits are required.

**Isotopic signatures of source regions**

The section on the isotopic signatures for identifying the predominant sources is interesting but needs some more work:

The Keeling plot technique ( $\delta^{13}$ C against 1/CH4) is only appropriate for a constant background. Some detail is required about how these Keeling plots were constructed. Is a constant background appropriate or are you plotting data over a period when there will be seasonal variability or interannual differences in the background? If you are plotting data over several years and the global background methane mole fraction and  $\delta^{13}$ C are changing then Keeling plots should not be used. If the constant background assumption cannot be made then Miller-Tans plots could be used instead to identify the source isotopic composition, e.g. Miller and Tans, 2003 https://doi.org/10.1034/j.1600-0889.2003.00020.x; Al-Shalan et al., 2022 https://doi.org/10.1016/j.atmosenv.2021.118763; Varga et al., 2021.

**Figure 9**

In (c) and (d) we see that source signatures for CS and KL increased in 2016-2020 compared with 2006-2010. This seems to contrast with line 13 on page 21 which talks of a decreasing trend in  $\delta^{13}$ C

What are the uncertainties in the trends in (e) and (f). Can you really say there is a trend?

**References**

The reference lists needs editing. Some of the references were missing from the reference list:

Watanabe et al., 2000; Remann et al., 2004; Remann et al., 2008; Li et al., 2017; Turnbull et al., 2015.

The reference Shuang-Xi Fang et al. (2013) should be deleted as this is already listed as Fang et al., 2013.

Kim et al., 2014 on page 2 should be Kim et al., 2015 to match the reference list.

**Other questions:**

As AMD is affected by local sources it would be helpful to use an inventory to suggest quantitatively what the anthropogenic emissions sources are (e.g. EDGAR, or UNFCCC) in the introduction.

Figure 1 – add a scale bar to this map.

What was the reason for drying the air rather than using the water correction built into Picarro software (see Rella et al., 2013)?

Page 2, line 31. A large ratio,  $CH_4/C_2H_6$  – explain what that means. Is high methane but low ethane indicative of a biogenic source?

Page 3 – lines 29/30. Were there 2 garbage incinerators or one? This part needs clarifying.

Page 7 - it's not clear how the filtering was applied to the data using HS, CD and MS. Are data outside of 1 s.d. of the mean filtered out?

Page 8 – line 15 – typo in HYSPLIT

Figure 7 – The growth rates for AMY, JGS and ULD aren't shown for 2018. I think this is because they are negative (-1 ppb) but this is still a result so they should be shown.

Why are some of the numbers in Table 4 written in bold?

Figure 8 – 2009 was unusual because there was no seasonal cycle – can you comment on why this was?

I didn't fully understand the PSS analysis – I think that needs some more detail.

---

## Author Comment (AC1)

**Author's responses to reviewer's comments follow. A copy of the reviewer comment is given (with comment 'number') followed by a response (blue font).**

**Response to referee 1**

1. General comment

The authors of this study present high-quality record of the atmospheric $CH_4$ mole fractions measured at three WMO/GAW stations in Korea: Anmyeondo (AMY), Jeju Gosan Suwolbong (JGS), and Ulleungdo (ULD). To confirm the data quality in detail, the authors evaluate the total measurement uncertainties at the three sites, which are less than GAW's compatibility goal for atmospheric $CH_4$ measurement (±2 ppb). Then, they investigate the characteristic features of the $CH_4$ variations by precisely analyzing the excess $CH_4$ mole fractions from the background variations, the diurnal cycles, the seasonal cycles, and the growth rates. From PSS analysis and the $^{13}CH_4$ isotopic data obtained at nearby site (TAP), they also discuss the temporal changes in the source region and the dominant source categories from a 5-year period of 2000-2006 to that of 2016-2020. Although I think the discussion based on the PSS and isotopic analyses have some problems, I believe that the data presented in this report are quite reliable and contribute to the global modeling studies especially for the $CH_4$ emission analysis from the east Asia.

I found that the paper contains material that should be published in Atmospheric Physics and Chemistry. However, I think that the authors should make more efforts to improve the manuscript as is suggested in followings before publication.

Thank you for your comments on the paper's value. We also appreciate your helpful comments to improve our manuscript. According to your specific comments and considerations, we revised our manuscript.

2. I'm not sure why the authors discuss in depth the influence of water vapor on the CRDS analyzer in Section 2.3 and 3.1. The CRDS analyzer provides dry $CH_4$ mole fraction by measuring the water vapor simultaneously. The study of Rella et al. (2013) revealed that the preliminary reduction of water vapor of sample air less than 1% allow CRDS analyzer to measure the $CH_4$ mole fraction within the level of GAW's compatibility goal (±2 ppb). The authors describe that the air samples are dried to −50C. I believe that the laboratory standard gases and working standard gases uses in this study are sufficiently dried (I think that the absolute dew points should be clarified in the text). In addition, if the air samples are dried by a cold trap of −50 C, the dilution effect is comparable to the reproducibility of the CRDS analyzer. Therefore, I think it would be better to simplify the description about $H_2O$ influence on the CRDS measurement.

Agree. But we do not use the water correction function provided by CRDS since we believe that $H_2O$ value also should be calibrated when we apply the correction algorism to our data and general water correction cannot be applied to each instrument in different environment. So we clarify this in the manuscript. Please see the answer of # 11.

Since WMO GAW community emphasize the importance of drying air and its bias, we would like to explain relevant information detailed to prove the quality of our data. Especially readers and data users can have questions about our working standards and its bias through propagating against laboratory standards. To remove this reservation, we dealt with our working standards very deeply in section 2.3.

On the other hand, though the measurement uncertainty from drying system is also negligible (0.006 to 0.008 ppb in Table 3), this value can be important. It is because that the standard gas is injected not through the drying system and separately from samples. However, the reviewer also mentioned the air dried at −50 C and the biases are very small. Therefore, it can be less valuable to deal with this deeply as one of uncertainty

factors. We would like to keep the explanations about working standards while simplifying about the measurement uncertainty.

*P4 L20:* Despite the Picarro provides built-in dry correction algorism, this is not applied to our data since generic water correction cannot be applied to each instrument in different environment. Though we dry our samples with this system, the biases resulted from the different $H_2O$ values between samples and standards or working and laboratory standards are considered here. This is described in section 2.3 and 3.1.

*P10 L25:* $U_{h2o}$ was computed from the differences in $H_2O$ (%) between the ambient airstream through the drying system and standard gases injected directly, bypassing the drying system. An ideal measurement would be through the analysis of the standard gases and air samples after they pass through the same drying system (WMO, 2016). However, our drying efficiency was not constant, so we injected standard gases directly as a reference value of $H_2O$. Therefore we considered the $CH_4$ dilution offsets between working standards and sample air while estimating the uncertainty with a similar method using Eq.(1) in section 2.3. Hourly $CH_4$ dilution maximum offsets are up to 0.009 ppb at AMY, 0.006 ppb at JGS and 0.009 ppb at ULD from 2016 to 2020. This uncertainty term was smallest (0.006 to 0.008 ppb) among all uncertainty factors in the KMA network as indicating the sampled air has negligible biases through our drying system

3. I'm very curious about the diurnal cycles of the L3 hourly data. If the data selection method described in Section 2.4.2 effectively remove the local effect, there is no statistically significant diurnal cycles in the L3 data. Such information would be useful to make the L3 data more reliable

We attached L3 hourly data through step 1 to 3 in 2020 for three stations here. The scale is different according to stations, but it can be confirmed that our selection method is reasonable. We also add these graphs in the supplementary as Figure S2 to demonstrate

our method is reasonable.

(a)

[Figure]

(b)

[Figure]

(c)

[Figure]

Figure A. The time series of hourly CH$_4$ data through our selection method from step 1 to

3 at (a) AMY (b) JGS and (c)ULD in 2020.

On the other hand, we also showed the L3 diurnal cycle at each station in 2020 here. As

reviewer mentioned the diurnal cycle is almost flat within ±2 ppb variations regardless of

stations. This variation might be resulted from the local meteorological effect by the radiation such as high boundary layer height in the daytime and stable in the night at least. Therefore, we have confidence that our L3 method remove local sink and source of CH₄.

(a)

[Figure]

Figure B. Mean diurnal variations of CH₄. Values show the average departure from the daily mean in each month at (a) AMY and (b) JGS from 2016 to 2020 and (c) at ULD in 2020.

4. I don't agree with the discussion based on the PSS and the $\delta^{13}CH_4$ analyses in Section 3.5. As for the PSS analysis, I'm not sure how to interpret the plots shown in Figs 9a and 9b. Although the authors seem to consider that the plots reflect the $CH_4$ flux distributions, the flux distribution is considerably different from the previously reported flux distributions (e.g., Ito et al., 2019). Additionally, I cannot believe that such large temporal change in the flux distribution occurred between the two periods. So, I think that the authors should clarify what the plots based on the PSS analysis mean. The authors only show the plots based on the observation at AMY. But I'm curious about the similar plots based on the data at JGS and ULD. If the distributions based on JGS and ULD are similar to Fig. 9b, such result would convince us to some extent that the PSS analysis is reliable. I'm not sure what the isotopic analysis means. The authors would like to suggest that influences of the biogenic sources increased from 2006-2010 to 2016-2020 because the $\delta^{13}C_{(CH4)}$ values for the latter period shows faster decreasing rates than those for the former period. However, the background $\delta^{13}C_{(CH4)}$ data also show the secularly decreasing trend (for example, you can see the recent $\delta^{13}C_{(CH4)}$ change in NIWA's home page: https://niwa.co.nz/atmosphere/our-data/trace-gas-plots/methane). So, how do the authors distinguish the regional influence on the $\delta^{13}C_{(CH4)}$ values from the background change? Additonally, why don't the authors use Miller-Tans plot to evaluate of the $\delta^{13}C$ values for the $CH_4$ regional sources? On the other hand, the values of the intercept of the Keeling plot for CS and KL increased from 2006-2010 to 2016-2020, suggesting the influence of the biogenic sources relatively decreased. This result is inconsistent with the above result. I'm also curious about the $\delta^{13}C_{(CH4)}$ data during the period from 2010 to 2016. Why don't the authors discuss the period from 2010 to 2016?

Thank you for the comment and we understand your considerations.

1) PSS analysis

   For the PSS analysis, we described the detailed information in #20 and revised the section 2.7 with detailed descriptions.

   When we analyze that ULD and JGS as well, the covered regions and $CH_4xs$ scale were different due to observatory location (Please see Supplementary Figure S4). However, we confirmed that they indicated reasonably same areas as major source regions such as southern China and western Korea from 2016 to 2020. Therefore, we have a high confidence about PSS analysis. We also reviewed the Ito et al.(2019). Bottom-up approach cannot be exactly matched to PSS analysis since PSS analysis calculate the air-mass residence time-weighted mean concentrations not the $CH_4$ flux as we mentioned in revised section 2.7. This is the same reason that each stations showed slightly different coverage for the source regions in PSS analysis. PSS analysis only can give information which area potentially affect the atmospheric $CH_4xs$ observed at each station.

   On the other hand, what we would like to emphasize is $CH_4$ growth rate was driven by mixed source in East Asia while global by biogenic sources mainly. Therefore, we used PSS analysis as a tool to figure out representative source regions in East Asia and implicated the $CH_4xs$ data from 2006 to 2020 (15 years). According to reviewer's suggestion, we analyzed Miller-Tans plots using the samples from source regions thorough PSS analysis every 5 years period to see the source changes.

2) Isotope analysis

   The reasons why we compare only between 2006/10 and 2016/2020, we would like to show the certain changes from end to end in a decade. According to your comments, we analyze the Miller-Tans plots every five years (2006/2010, 2011/2015, and 2016/2020) and could find out clear changing in CN and CS region. We showed all graph in Supplementary Figure S7. According to the reanalysis, we revised all manuscripts in section 3.5.

*Section 3.5 from P21 L26:* To understand the source regions affected AMY CH$_4$ level, we analysed PSS with hourly CH$_4$xs from 2006 to 2020. CH$_4$xs did not vary much and was 49±74 ppb during 2006–2010 and 50±70 ppb during 2016–2020. According to the PSS analysis, affecting major source regions were CN, CS and KL sectors (Fig. 9 (a)). Sources affecting CS and KL are paddy and livestock fields and that for CN was reported to be fossil fuel emissions mainly (Zhang et al., 2011, Ito et al., 2022, Chen et al., 2022).

Through the HYSPLIT cluster analysis from 2006 to 2020, we categorized the TAP $\delta^{13}C_{(CH4)}$ data and select the samples only affected by each source regions, CN, CS, and KL, respectively (section 2.6). Using TAP $\delta^{13}C_{(CH4)}$ long-term data from 2006 to 2020 affected by CN, CS and KL, Miller-Tans plots indicated that emissions from CN were mainly related to fossil fuel or biomass burning (–44.3±1.8‰), while CS (–56.1±1.5‰) and KL (– 54.6±1.2‰) were affected more by biogenic sources during 2006–2020 (Fig.9 (b)). Sherwood et al. (2017) reported unweighted global mean $\delta^{13}C$ of –44.8±10.7‰ from fossil fuel use, –26.2±4‰ from biomass burning, and –61.7±6.2‰ from microbial sources. Even though the uncertainty of isotopic source signature is quite large, CH$_4$ formed at high temperature such as combustion is enriched in the heavier isotope while CH$_4$ from wetland, rice paddies and livestock is depleted. Therefore, our isotope analysis was well matched to reported source regions.

On the other hand, isotope signatures were shifted slightly in China (CN and CS) while for Korea (KL) it was steady in the uncertainty range from 2006 to 2020. When we analyze the Miller-Tans plots in every 5 years (Fig. S7), for CN the slope was -38±3‰ in 2006/10 but it became depleted -45±2.4‰ in 2016/20 while those value was enriched from -59.8±1.5‰ to -51.9±2.5‰ in CS. KL showed the quite constant values from -55 to -54‰ in the same period. This suggested that CH$_4$ growth rate in East Asia was affected not only biogenic but also pyrogenic sources, unlike global. The recent global accelerated increase in atmospheric CH$_4$ was more related to biogenic sources such as agriculture and wetland (Jackson et al., 2020, Lan et al., 2021).

Since the $CH_4$ emissions from agriculture and livestock accounted for 30% and 36% in China and Korea respectively in 2020 (Crippa et al., 2022), $CH_4$ might be increased by temperature impacts on biogenic $CH_4$ source. However, the fast urbanization and energy consumption strategy also can affect these regions. Especially the coal emissions decreased from 2010 in China (Liu et al., 2021) but the coal to gas policy lead natural gas consumptions increase again in China (Wang et al., 2022).

Overall, AMY and global growth rates were renewed in 2006 and during 2006–2020; the increasing trend could be linked to mixed biogenic and fossil fuel sources in East Asia while global to more biogenic sources.

Regarding this, we revised the abstract and section 4. Summary and conclusion as well.

*P1 L2:* From the long-term records at AMY, we confirmed that growth rate increased 3.3 ppb·yr$^{-1}$ during 2006/2010 and by 8.3 ppb·yr$^{-1}$ from 2016 to 2020, which is similar trend to global. It is reported that the recent global accelerated $CH_4$ growth rate was related to biogenic sources. However, isotopic signature using $\delta^{13}CH_4$ explained that $CH_4$ sources are becoming mixture of not only biogenic but also fossil fuel sources in East Asia from 2006 to 2020. We confirmed that long-term high-quality data can help understand changes in $CH_4$ emissions in East Asia.

*P24 L10:* From the long-term analysis of $CH_4$ data at AMY, average $CH_4$ growth rate was 3.3 ppb·yr$^{-1}$ during 2006–2010, but increased to 8.3 ppb·yr$^{-1}$ in 2016–2020 as similar to the global trend. Through the source distributions with our PSS analysis using $CH_4$xs data, CN, CS and KL sectors were main regions to affect atmospheric $CH_4$ observed at AMY. Isotope signature based on Miller-Tans plots at CN represents fossil fuel or burning activities while CS and KL biogenic sources during 2006-2020. However, we infer atmospheric $CH_4$ drivers changes in air masses arriving from China sector, CN and CS. For East Asia the increasing trend could be linked to

mixed biogenic and fossil fuel sources while global to more biogenic sources (e.g. agriculture and wetland). Through this study, we confirmed that long-term high-quality data can help understand changes in $CH_4$ emissions in East Asia. Also, further studies are necessary based on observations to understand sources changes in East Asia since there is a discrepancy between reported inventory and observations (Wang et al., 2022).

**Specific comments:**

5.  Page 2, line 28: "Kim et al., 2014" should be "Kim et al., 2015".

Corrected

6.  Page 2, line 30-32: "Li et al., 2020" and "Li et al., 2022" are not listed in References.

Corrected

7.  Page 2, line 36: "Turnbull et al., 2015" is not listed in References.

Corrected

8.  Page 3, line3: "Watanabe et al. 2000" is not listed in References.

Corrected

9.  Page 4, line 17: What's the "−90°C" mean? If it's not the trap temperature, it should be removed to avoid confusion.

Corrected

10. Page 5, line 6: I believe that the four laboratory standards are kept in high-pressure cylinders. I think that it would be better to clarify what kind of cylinders are used for the laboratory standards (volume and material).

We added the information here.

*P5, L11 :* They are provided with 29.5 L aluminum cylinder (Luxfer, UK) by CCL and filled with 130 bar (https://gml.noaa.gov/ccl/service.html, last access: 5 Jan.,2023). Their recalibration in every 3 years is highly recommended for $CO_2$, $CH_4$ and CO. Our laboratory standards have been recalibrated every 3 years or replaced to new sets since the first using in 2012 according to the recommendation but remaining pressure in each cylinder is still high around 100 bar.

11. Page 5, line 13-20: I'm a little bit confused by this part. I think that the CRDS analyzer (Picarro) give dry mol fractions of the atmospheric $CH_4$. So, why does the authors discuss the dilution effect of water vapor?

We do not use the water correction function provided by CRDS since we believe that $H_2O$ value also should be calibrated when we apply the correction algorism to our data and general water correction cannot be applied to each instrument in different environment. In this context, many networks and stations don't use dry mole fraction reported by Picarro and apply post-processing water correction (Hazan et al., 2016, Zellweger et al., 2016).

In our case, we don't use post-processing since we dry our samples through cryogenic method (detailed in section 2.2). Also, even when we compress air into cylinder for working standards, we dry them. However, it should be considered the bias resulted from the differences of $H_2O$ values between samples and standards or working and laboratory standards. To avoid the confusion, we added the sentence.

*P 4, L20:* Despite the Picarro provides built-in dry correction algorism, this is not applied to our data since generic water correction cannot be applied to each instrument in different environment. Though we dry our samples with this system, the biases resulted from the different $H_2O$ values between samples and standards or working and laboratory standards are considered here. This is described in section 2.3 and 3.1.

[Reference]

Hazan, L., J. Tarniewicz, M. Ramonet, O. Laurent, and A. Abbaris.: Automatic processing of atmospheric CO2 and CH4 mole frations at the ICOS Atmosphere Thematic Centre, Atmos. Meas.Tech., 9, 4719-4736, 2016

Zellweger, C., L. Emmenegger, M. Firdaus, J. Hatakka, M. Hemann, E. Kozlova, T. G. Spain, M. Steinbacher, M. V. van der Schoot, B. Buchmann.: Assessment of recent advances in measurement techniques for atmospheric carbon dioxide and methane observations, Atmos. Meas.Tech., 9, 4737-4757, 2016

12. Page 5, line 17 (Eq. 1): It should be better to clarify what the $C_{dilution}$ and $C_{dry}$ mean. And the definition of $H_{act}$, "$H_2O$ difference between standard gases and samples", is rather confusing. Is it "$H_2O$ difference between laboratory standard gases and working standard gases"?

Agree. We corrected the sentence and added the definition.

*P6, L7* where *C* is the $CH_4$ mole fraction and $H_{act}$ is the actual water mole fraction (in %). Here $H_{act}$ is the $H_2O$ difference between laboratory and working standards. $C_{dry}$ is the $CH_4$ mole fraction of laboratory standard while $C_{dilution}$ is the expected $CH_4$ mole fraction when $H_{act}$ existed. We calculated the bias from the difference $C_{dry}$ and $C_{dilution}$.

13. Page 7, line 10-11: It should be better to clarify the detail of the fitting function (order of polynomial and order of harmonics) and the cut-off frequency for the digital filtering based on Thoning et al. (1989).

To clarify the method, we add the explanation in Appendix B, Supplementary (and like below).

*P7, L28:* The details were described in Appendix B the supplementary.

APPENDIX B in Supplementary

After the baseline selection through the 3 steps (section 2.4.2), selected data are computed by curve-fitting methods. The basic components of the curve are a second-order polynomial representing the long-term trend, and a series of harmonics representing the average seasonal cycles (Eq.S1).

$$f(t) = a_0 + a_1 t + a_2 t^2 + \Sigma_{k=1}^{4}\left[ b_{2k-1}\sin\left(\frac{2\pi kt}{T}\right) + b_{2k}\cos\left(\frac{2\pi kt}{T}\right)\right]$$

Equation S1. Function fit

Where t is the time in days and T is 365.25 days.

And then calculate and filter the residuals of the selected daily data about the fit function. The method used is to transform the residuals into the frequency domain using a Fast Fourier Transform (FFT), apply a low pass filter function (Eq. 2) to the frequency data, and then transform the filtered data to the real domain using an inverse FFT.

$$H(f) = \exp\left[-\ln(2) * \left(\frac{f}{f_c}\right)^6\right]$$

Equation S2. Low pass filter function

Where f is frequency and fc is cut-off frequency. Here fc is cut-off values of 80 days for the short-term filter and 667 days for the long-term filter.

Finally determine the smoothed curve of interest by combining the function with the filtered data.

14. Page 7, line 12: It should be better to clarify how to compute the monthly data, long-term trend, and seasonal amplitude in detail.

The definitions are derived from Global Monitoring Laboratory - Carbon Cycle Greenhouse Gases (noaa.gov) (https://gml.noaa.gov/ccgg/mbl/crvfit/crvfit.html, last access, 16, 2023). These explanations are also included in Appendix C, the supplementary (like below).

*P7, L29*: Finally, we can get the L3 monthly data, long-term trend and seasonal amplitude after applying Thoning et al. (1989). The detailed definitions are in Appendix C, supplementary.

APPENDIX C in Supplementary

The definitions are derived from Global Monitoring Laboratory - Carbon Cycle Greenhouse Gases (noaa.gov) (https://gml.noaa.gov/ccgg/mbl/crvfit/crvfit.html, last access, 16, 2023).

Simply L3 monthly mean are calculated by average of daily smoothed data from Eq. S1 and Eq. S2. Daily smoothed data is the function fit plus the filtered residuals using the short term cut off (80 day/cycle)-1.

Long-term trend is upward growth in the data with the seasonal cycle removed. This is the polynomial part of the function fit plus the filtered residuals using the long-term cut-off value with (667 day/cycle)-1.

Seasonal amplitude is magnitude of the peak to trough of the detrended seasonal cycle. Detrended seasonal cycle can be obtained by subtracting the trend curve from the smooth curve.

15. Page 7, Table 2: There are no values of MS for AMY and ULD. Does it mean that MS criteria were not applied to the data for AMY and ULD?

Sorry for the confusion. The same value of MS is applied to AMY and ULD as well. We put the explanation like:

*Table 2: $1.8\sigma_{30d}$ (for all three stations)*

*P7, L21:* C is the standard deviation of 30 days moving average multiplied by $\alpha$ and here $1.8\sigma_{30d}$ is applied to all three stations as C.

16. Page 7, line 25-26: The geological locations of WLG and RYO are already given in page 3, line 1-3.

We deleted the explanation and used only abbreviation.

17. Page 8, line 6-8: It should be better to add the reference for the $CH_4$ isotopic measurement at INSTAAR.

We added the reference in the manuscript on page 8 line 19.

[Refence]

Miller, J. B., K. A. Mack, R. Dissly, J. W. C. White, E. J. Dlugokecky, P. P. Tans.: Development of analytical methods and measurements of 13C/12C in atmospheric $CH_4$ from the NOAA Climate Monitoring and Diagnostics Laboratory Global Air Sampling Network, J. Geophys. Res. Atmos. 107, https://doi.org/10.1029/2001JD000630, 2002

18. Page 8, line 5 and 9: It says here that the distance between TAP and AMY is 24 km. But other parts in this manuscript, the distance is 28 km. Which value is correct?

Thank you for the correction. We revised distance between TAP and AMY to 28 km through all manuscript.

19. Page 8, line 9-10: It should be better to add a figure showing synoptic scale variations of $CH_4$ observed at AMY and TAP.

We added the relevant figures in supplementary as Figure S3 and the reference in the manuscript on page 8 line 21.

*P8, L21:* Since TAP and AMY are only 28 km apart, their data are representative of the same region under large synoptic conditions (Fig. S3), especially for well mixed air. These data were thus used to trace the changes in the surrounding environment in East Asia (section 3.5).

20. Page 8, line 35-page 9, line 4 (2.7 PSS analysis): Since I'm not sure PSS analysis, I tried to read the cited papers, Remann et al., 2004, 2008; Li et al., 2017. But these papers are not listed in References. If the single line for each backward trajectory is used for the calculation of $C_{(i,j)}$, the sensitivity of the potential source strength increase with the distance from the start point in comparison with the reality. This effect would result in the overestimation of the potential source strength with increasing the distance from the station, wouldn't it?

I agree. As you mentioned, the potential source strength can be overestimated around the receptor site because the more trajectories pass through the grid cell i,j, as it more close to receptor site. In order to correct the overestimations, we adapted the geometric adjustment factors (Fi,j) as described by Poirot and Wishinski (1986). Calculation of adjusted residence time over each grid cell of i,j was accomplished using the following two equations:

$$T_{(i,j,a)} = \sum_{n=1}^{N} \frac{S_{(i,j,n,a)}}{V_{(n,a)}} * F_{(i,j)} \qquad Eq\ (1)$$

$$F_{(i,j)} = \frac{\pi\left[\left(D_{(i,j)} + \frac{d_g}{2}\right)^2 - \left(D_{(i,j)} - \frac{d_g}{2}\right)^2\right]}{d_g \times d_g} \qquad Eq\ (2)$$

As shown in below Illustrations of residence time calculation for a single randomly selected back-trajectory (Figure A), S(i,j,n,a) is the length of that portion of the nth segment of the a back-trajectory which falls over grid cell i,j. V(n,a) is the average speed of the air parcel as it travels along the nth segment of the a back-trajectory. The geometric adjustment factor (Fi,j) were calculated using Equation (2). Where dg is the length of the grid cell of i,j; D(i,j) is the distance from receptor site to the grid cell of i,j.

[Figure]

Figure A. Illustration of residence time calculation on a grid cell (i,j) of a HYSPLIT back-trajectory arrived at Jeju Gosan station.

Additionally we added missing reference and revised the manuscript like below:

*Section 2.7 Potential Source Strength (PSS) analysis*

To identify and illustrate the potential source distributions for regional pollutions, we calculated the PSS using the trajectory statistics approach, which has often been applied to estimate the potential source areas of greenhouse gases (Reimann et al., 2004, 2008; Li et al., 2017). The trajectory statistics approach was introduced first by Seibert et al. (1994). The underlying assumption of the method is that elevated atmospheric levels at an observation site are proportially related to the air mass residence time on a specific grid cell over which observed air mass has been passing. Thus, this method simply calculates the airmass residence time weighted mean concentrations (here mole fractions) for target compounds ($CH_4$ in this study) in the domain with $0.5 \times 0.5$ grids using the following formula (Eq. 2):

$$\overline{C_{(i,j)}} = \frac{\sum_{a=1}^{M} T_{(i,j,a)} C_a}{\sum_{a=1}^{M} T_{(i,j,a)}} \qquad \text{(Eq. 2)}$$

where $C_{(i, j)}$ represents the potential source strength of the grid cell i, j as a potential source region of the target compound ($CH_4$); a is the index of the trajectory; M is the total number of trajectories that passed through cell i, j; Ca is the enhanced mole fraction (difference from background mole fractions mentioned in section 3.2 below) measured during the arrival of trajectory a; and $T_{(i,j,a)}$ is the residence time of trajectory a spent over grid cell i, j and which were calculated using the method described by Poirot and Wishinski (1986) as following formula (Eq.3)

$$T_{(i,j,a)} = \sum_{n=1}^{N} \frac{S_{(i,j,n,a)}}{V_{(n,a)}} \qquad \text{(Eq. 3)}$$

where $S_{(i,j,n,a)}$ is the length of that portion of the $n^{th}$ segment of the a back-trajectory which falls over grid cell i,j. $V_{(n,a)}$ is the average speed of the air parcel as it travels along the $n^{th}$ segment of the a back-trajectory. Backward trajectories were calculated using the HYSPLIT model of the NOAA Air Resources Laboratory (ARL) using meteorological information from the Global Data Assimilation System (GDAS) model. For the trajectory reliability, we used only the 4-days (96h) backward trajectories at an altitude of 500 m above the mean sea level. To consider the influence of air masses on emissions at ground level, the air masses passing above the boundary layer height (BLH) were excluded. The BLH can was obtained from the HYSPLIT model. To exclude the influences of emission sources surrounding AMY stations, enhanced $CH_4$ data with wind speeds lower than 2 m·s$^{-1}$ were ruled out for the PSS analysis. When we compare PSS analysis among AMY, JGS, and ULD using $CH_4$xs data from 2016 to 2020, they showed the similar source regions while the coverage and $CH_4$xs are slightly different (Fig. S4.).

[Reference]

Poirot, R L, and Wishinski, P R. Visibility, sulfate and air mass history associated with the summertime aerosol in northern Vermont. Atmos. Environ., 20, 1457-1469, 1986

21. Page 12, line 8-9: Are the characteristics of the bivariate polar plots for the three stations (Figs. 3-5) same with those for the other years?

Yes. The reason why we showed the bivariate plots in 2018 is there was only weak typhoon, so the year of 2018 is very reasonable. We showed the example in 2019 here with sever typhoon but basically all plots show the similar characteristics which we described in the manuscript. We also revised sentence like below:

*P13, L8:* To understand the influence of local surface wind on observed $CH_4$, bivariate polar plots were used for 2018, the year less affected by typhoons compared to other years

[Figure]

Figure A. Bivariate polar plots for observed $CH_4$ (L2 hourly) in spring (a), summer (b), autumn (c), and winter (d) at AMY in 2019.

[Figure]

Figure B. Bivariate polar plots for observed CH$_4$ (L2 hourly) in spring (a), summer (b), autumn (c), and winter (d) at JGS in 2019.

[Figure]

Figure C. Bivariate polar plots for observed CH$_4$ (L2 hourly) in spring (a), summer (b), autumn (c), and winter (d) at ULD in 2019.

22. Page 18, line 10-11: The seasonal cycle of the atmospheric CH₄ is also influenced by the atmospheric transport.

Agree. We revised the sentence here.

*P19, L10:* The seasonal amplitude is related to the seasonal atmospheric transport, the combination of CH₄ surface flux distribution and chemical loss by reactions with OH and by soil loss.

23. Page 18, line 19: The distance between TAP and AMY is repeatedly described in this manuscript.

Corrected.

24. Page 18, line 20: "during" what?

Corrected

25. Page 19, Figure 7b: Why don't the authors plot the negative growth rate in the figure?

We revised to a new graph.

26. Page 19, line 12: Was the annual growth rate calculated from the annual means based on the monthly means, listed in Table 4? If so, it would be helpful for the readers to describe that in the manuscript.

It was described in Table 4 caption, but we also explain this in section 2.4.2.

*P8 L13:* Annual means are averaged by monthly means while annual growth are derived from the difference of consecutive annual means.

27. Page 19, line 15: CH$_4$ at AMY, JGS, JLD, and RYO showed significant increases from 2016 to 2017, but small (or negative) increases from 2017 to 2018.

Corrected

*P 21 L1*: Especially from 2016 to 2017, WLG showed no increase in CH$_4$, while CH$_4$ at other stations showed small or negative increases from 2017 to 2018.

28. Page 20, line 1: There is relatively large differences in the CH$_4$ growth rate between WLG and the WMO global mean in 2017 and 2020.

Even though they showed the value difference, the increase/decrease trend is similar. According to regions, the value of station's growth rate can be different, but the trend can be similar when they capture the baseline conditions. The growth rate value can be differed from the regions since their source and sink distributions are different at each region.

We revised the sentence to avoid confusion.

*P 21 L2:* Normally the growth rate in CH$_4$ at WLG matches well with the WMO global increase/decrease trend (Fig. 7. (b)).

29. Page 20, line 5-6: There is no description about the $\delta^{13}CH_4$ data at AMY. Please add the explanations for the data. In addition, since the plot shown in Fig. S3 is not Keeling plot but Miller-Tans plot, the slopes of the plots represent the $\delta^{13}C$ of the $CH_4$ sources.

Agree. We added AMY flask explanations.

*P8 L22:* AMY started flask sampling for $CH_4$ and $\delta^{13}C_{(CH4)}$ in December 2013 with same method like TAP and these data were used only for characterization of $CH_4$ at AMY in section 3.5.

It is typo so that we corrected it in the manuscript and the Fig. S3 (now it is Fig.S6) as well.

*P21 L6:* Miller-Tans plots shows the signature of $CH_4$ increments into background air of Mauna Loa (Fig. S6). And the slope was –52.3±2.2‰ in winter and –53.7±0.7‰ in summer at AMY during 2016 to 2020..

30. Page 21, line 1: "Sources affecting CS and KL were paddy fields and ..." ?

We wrote below:

*P21 L28:* Sources affecting CS and KL are paddy and livestock fields and that for CN was reported to be fossil fuel emissions mainly (Zhang et al., 2011, Ito et al., 2022, Chen et al., 2022).

31. Page 21, line 13: I'm not sure what "this assumption" means.

We revised whole descriptions in section 3.5. as we described in #4 above.

32. Page 21 line 3-14: Are all the flask data collected at TAP plotted in Figs 9? Are the source regions of all the flask data classified into three sectors, CS, CN, and KL? How are they classified?

To avoid the confusion, we added detailed information.

*P22 L1:* Through the HYSPLIT cluster analysis from 2006 to 2020, we categorized the TAP $\delta^{13}C_{(CH4)}$ data and selected the samples only affected by each source regions, CN, CS, and KL, respectively (section 2.6).

33. Page 26, line 21-23: "Shuang-Xi et al., 2013" is not cited in the text.

corrected

34. Page 27, line 3-5: "Winderlich et al., 2010" is not cited in the text.

It was cited in p 12 line 10

---

## Author Comment (AC2)

**Author's responses to reviewer's comments follow. A copy of the reviewer comment is given (with comment 'number') followed by a response (blue font).**

**Response to referee 2**

1. General comment

The clear explanation of the set-up of the measurements at the 3 stations, scale propagation and uncertainty would be valuable to others developing and refining greenhouse gas measurement systems. The publication of the high-quality data from the stations, and discussion of the regional context is good. I think further work should be done on the source identification (section 3.5). This may change the overall conclusion of the work – i.e. whether the changes in the region are a relative increase in biogenic sources or not. Before publication some edits are required.

Thank you for your comments on the paper's value. We also appreciate your helpful comments to improve our manuscript. According to your specific comments, we revised our manuscript, especially isotope analysis results. Also, we tried to explain PSS analysis more detailed.

2. Isotopic signatures of source regions

The section on the isotopic signatures for identifying the predominant sources is interesting but needs some more work: The Keeling plot technique ($\delta$ 13C against 1/CH4) is only appropriate for a constant background. Some detail is required about how these Keeling plots were constructed. Is a constant background appropriate or are you plotting data over a period when there will be seasonal variability or interannual differences in the background? If you are plotting data over several years and the global background methane mole fraction and $\delta$ 13C are changing, then Keeling plots should not be used. If the constant background assumption cannot be made then Miller-Tans plots could be used instead to identify the source isotopic

composition, e.g. Miller and Tans, 2003 https://doi.org/10.1034/j.1600-0889.2003.00020.x; Al-Shalan et al., 2022 https://doi.org/10.1016/j.atmosenv.2021.118763; Varga et al., 2021. Figure 9 In (c) and (d) we see that source signatures for CS and KL increased in 2016-2020 compared with 2006-2010. This seems to contrast with line 13 on page 21 which talks of a decreasing trend in δ 13C What are the uncertainties in the trends in (e) and (f). Can you really say there is a trend?

Agree. According to your comments, we analysed the Miller-Tans plot and could get more clear results for source changes. So we use PSS analysis as a tool to show the representative area where affected atmospheric $CH_4$ at AMY. Through the HYSPLIT cluster analysis based on TAP flask sampling dates, we selected the samples from the source regions where PSS analysis indicated. Miller-Tans plots were analysed every five years to see its changes.

Finally, we revised whole description in section 3.5. like below and we also added new graph in the manuscript and supplementary:

*Section 3.5 from P21 L26:* To understand the source regions affected AMY $CH_4$ level, we analysed PSS with hourly $CH_4xs$ from 2006 to 2020. $CH_4xs$ did not vary much and was $49\pm74$ ppb during 2006–2010 and $50\pm70$ ppb during 2016–2020. According to the PSS analysis, affecting major source regions were CN, CS and KL sectors (Fig. 9 (a)). Sources affecting CS and KL are paddy and livestock fields and that for CN was reported to be fossil fuel emissions mainly (Zhang et al., 2011, Ito et al., 2022, Chen et al., 2022).

Through the HYSPLIT cluster analysis from 2006 to 2020, we categorized the TAP $\delta^{13}C_{(CH4)}$ data and select the samples only affected by each source regions, CN, CS, and KL, respectively

(section 2.6). Using TAP $\delta^{13}C_{(CH4)}$ long-term data from 2006 to 2020 affected by CN, CS and KL, Miller-Tans plots indicated that emissions from CN were mainly related to fossil fuel or biomass burning ($-44.3\pm1.8‰$), while CS ($-56.1\pm1.5‰$) and KL ($-54.6\pm1.2‰$) were affected more by biogenic sources during 2006–2020 (Fig.9 (b)). Sherwood et al. (2017) reported unweighted global mean $\delta^{13}C$ of $-44.8\pm10.7‰$ from fossil fuel use, $-26.2\pm4‰$ from biomass burning, and $-61.7\pm6.2‰$ from microbial sources. Even though the uncertainty of isotopic source signature is quite large, $CH_4$ formed at high temperature such as combustion is enriched in the heavier isotope while $CH_4$ from wetland, rice paddies and livestock is depleted. Therefore, our isotope analysis was well matched to reported source regions.

On the other hand, isotope signatures were shifted slightly in China (CN and CS) while for Korea (KL) it was steady in the uncertainty range from 2006 to 2020. When we analyze the Miller-Tans plots in every 5 years (Fig. S7), for CN the slope was $-38\pm3‰$ in 2006/10 but it became depleted $-45\pm2.4‰$ in 2016/20 while those value was enriched from $-59.8\pm1.5‰$ to $-51.9\pm2.5‰$ in CS. KL showed the quite constant values from $-55$ to $-54‰$ in the same period.

This suggested that $CH_4$ growth rate in East Asia was affected not only biogenic but also pyrogenic sources, unlike global. The recent global accelerated increase in atmospheric $CH_4$ was more related to biogenic sources such as agriculture and wetland (Jackson et al., 2020, Lan et al., 2021).

Since the $CH_4$ emissions from agriculture and livestock accounted for 30% and 36% in China and Korea respectively in 2020 (Crippa et al., 2022), $CH_4$ might be increased by temperature impacts on biogenic $CH_4$ source. However, the fast urbanization and energy consumption strategy also can affect these regions. Especially the coal emissions decreased from 2010 in China (Liu et al., 2021) but the coal to gas policy lead natural gas consumptions increase again in China (Wang et al., 2022).

Overall, AMY and global growth rates were renewed in 2006 and during 2006–2020; the increasing trend could be linked to mixed biogenic and fossil fuel sources in East Asia while

global to more biogenic sources.

Regarding this, we revised the abstract and section 4. Summary and conclusion as well.

*P1 L2:* From the long-term records at AMY, we confirmed that growth rate increased 3.3 ppb·yr[-1] during 2006/2010 and by 8.3 ppb·yr[-1] from 2016 to 2020, which is similar trend to global. It is reported that the recent global accelerated $CH_4$ growth rate was related to biogenic sources. However, isotopic signature using $\delta^{13}CH_4$ explained that $CH_4$ sources are becoming mixture of not only biogenic but also fossil fuel sources in East Asia from 2006 to 2020. We confirmed that long-term high-quality data can help understand changes in $CH_4$ emissions in East Asia.

*P24 L10:* From the long-term analysis of $CH_4$ data at AMY, average $CH_4$ growth rate was 3.3 ppb·yr[-1] during 2006–2010, but increased to 8.3 ppb·yr[-1] in 2016–2020 as similar to the global trend. Through the source distributions with our PSS analysis using $CH_4$xs data, CN, CS and KL sectors were main regions to affect atmospheric $CH_4$ observed at AMY. Isotope signature based on Miller-Tans plots at CN represents fossil fuel or burning activities while CS and KL biogenic sources during 2006-2020. However, we infer atmospheric $CH_4$ drivers changes in air masses arriving from China sector, CN and CS. For East Asia the increasing trend could be linked to mixed biogenic and fossil fuel sources while global to more biogenic sources (e.g. agriculture and wetland). Through this study, we confirmed that long-term high-quality data can help understand changes in $CH_4$ emissions in East Asia. Also, further studies are necessary based on observations to understand sources changes in East Asia since there is a discrepancy between reported inventory and observations (Wang et al., 2022).

3. References

The reference lists need editing. Some of the references were missing from the reference list: Watanabe et al., 2000; Remann et al., 2004; Remann et al., 2008; Li et al., 2017; Turnbull et al., 2015. The reference Shuang-Xi Fang et al. (2013) should be deleted as this is already listed as Fang et al., 2013. Kim et al., 2014 on page 2 should be Kim et al., 2015 to match the reference list.

Corrected

4. Other questions:

4.1 As AMD is affected by local sources it would be helpful to use an inventory to suggest quantitatively what the anthropogenic emissions sources are (e.g. EDGAR, or UNFCCC) in the introduction.

We added the information of anthropogenic source in Korea in the introductions with the reference of EDGAR.

*P 2, L10:* China has also the largest anthropogenic $CH_4$ emissions in the world mainly from solid fuel (34%), rice cultivations (20%) and enteric fermentation (10%), respectively (Janssens-Maenhout et al., 2019; Crippa et al., 2022).
*P 2, L14:* South Korea major $CH_4$ emissions are derived from wastewater treatment (40%), enteric fermentation (22%) and then rice cultivations (14%) respectively (Crippa et al., 2022).

4.2 Figure 1 – add a scale bar to this map.

Corrected

4.3 What was the reason for drying the air rather than using the water correction built into Picarro software (see Rella et al., 2013)?

We do not use the water correction function provided by CRDS since we believe that $H_2O$ value also should be calibrated when we apply the correction algorism to our data and general water correction cannot be applied to each instrument in different environment. In this context, many networks and stations don't use dry mole fraction reported by Picarro and apply post-processing water correction (Hazan et al., 2016, Zellweger et al., 2016).

In our case, we don't use post-processing since we dry our samples through cryogenic method (detailed in section 2.2). Also, even when we compress air into cylinder for working standards, we dry them. However, it should be considered the bias resulted from the differences of $H_2O$ values between samples and standards or working and laboratory standards. To avoid the confusion, we added the sentence.

*P 4, L20:* Despite the Picarro provides built-in dry correction algorism, this is not applied to our data since generic water correction cannot be applied to each instrument in different environment. Though we dry our samples with this system, the biases resulted from the different $H_2O$ values between samples and standards or working and laboratory standards are considered here. This is described in section 2.3 and 3.1.

[Reference]

Hazan, L., J. Tarniewicz, M. Ramonet, O. Laurent, and A. Abbaris.: Automatic processing of atmospheric CO2 and CH4 mole frations at the ICOS Atmosphere Thematic Centre, Atmos. Meas.Tech., 9, 4719-4736, 2016

Zellweger, C., L. Emmenegger, M. Firdaus, J. Hatakka, M. Hemann, E. Kozlova, T. G. Spain, M. Steinbacher, M. V. van der Schoot, B. Buchmann.: Assessment of recent advances in measurement techniques for atmospheric carbon dioxide and methane observations, Atmos. Meas.Tech., 9, 4737-4757, 2016

4.4 Page 2, line 31. A large ratio, CH4/C2H6 – explain what that means. Is high methane but low ethane indicative of a biogenic source?

We explicitly explained what it means.

*P2 L32*: The ratio, $CH_4/C_2H_6$, was observed 53 ppb·ppb$^{-1}$ during KORUS-AQ campaign from May to June 2016 which seems to be associated fossil fuel in Seoul and Busan while it was 150 to 250 ppb·ppb$^{-1}$ related to biogenic emissions such as rice paddies in southern western part of South Korea (Li et al., 2022).

4.5 Page 3 – lines 29/30. Were there 2 garbage incinerators or one? This part needs clarifying.

Thank you for the correction. We revised the sentence.

*P3 L31:* In the southwestern area, there is a small brickyard 200 m from the station and a garbage incinerator within 100 m. The garbage incineration facility was moved to the north side of island in December 2016.

4.6 Page 7 – it's not clear how the filtering was applied to the data using HS, CD and MS. Are data outside of 1 s.d. of the mean filtered out?

Since this method was published in Seo et al.(2021), we did not handle this process precisely. However, we tried to explain our method with more clear ways here to avoid the confusions.

*P7 L14* There are three steps to select the background levels (L3 hourly data) from valid L2 hourly data:

Step 1) HS(t) ≤ A

Step 2) $|HA(t) - HA(t-1)| \leq B$ or $|HA(t) - HA(t+1)| \leq B$

Step 3) $|HA(t) - 30$ days moving median of of HA $| \leq C$.

Where HS represents $CH_4$ hourly standard deviation, HA is $CH_4$ hourly means and t represents time in hours. In step 3), t is the middle of the time window. A, B and C are criteria determined empirically for each step, as given in Table 2. C is the standard deviation of 30 days moving average multiplied by $\alpha$ and here $1.8\sigma_{30d}$ is applied to all three stations as C. Even though the data were selected by step 1) and 2), high $CH_4$ levels remained because of long-lasting stagnant conditions (e.g. over 6 days). Therefore, we also apply step 3). This process retained 21–52% of the data at each station, which were defined as L3 hourly on observations (Fig. S2). To get L3 daily/monthly data, the method developed by Thoning et al. (1989) was used to fit smooth curves to the daily averages computed by L3 hourly data. The methods reduce noise induced by synoptic-scale atmospheric variability, fill measurement gaps, and are used to represent the regional baseline. The details were described in Appendix B the supplementary. Finally, we can get the L3 daily data, L3 monthly data, long-term trend and seasonal amplitude after applying Thoning et al. (1989). The detailed definitions are in Appendix C, supplementary.

We also revised the table 2 and added the supplementary figure (Figure S2) to show our method.

Table 2. Criteria and percentage of selected background levels from observed data at each station.

| Station ID | AMY | JGS | ULD |
|---|---|---|---|
| Data period | 1999 to 2020 | 2012 to 2020 | 2012 to 2020 |
| A [ppb] | 2.1 | 2.1 | 2.8 |
| B [ppb] | 4.9 | 5.2 | 3.6 |

| C [ppb] | | $1.8\sigma_{30d}$ (for all three stations) | |
|---|---|---|---|
| Spring, MAM [%] | 29.1 | 46.6 | 57.9 |
| Summer, JJA [%] | 11.0 | 33.5 | 37.6 |
| Autumn, SON [%] | 16.9 | 30.9 | 53.2 |
| Winter, DJF [%] | 28.4 | 49.1 | 58.9 |
| Total [%] | 21.3 | 40.64 | 52.2 |

(a)

[Figure]

(b)

[Figure]

(c)

[Figure]

Figure S2. The time series of hourly CH₄ data through our selection method from step 1 to 3 at (a) AMY (b) JGS and (c)ULD in 2020.

4.7 Page 8 – line 15 – typo in HYSPLIT

Corrected

4.8 Figure 7 – The growth rates for AMY, JGS and ULD aren't shown for 2018. I think this is because they are negative (-1 ppb) but this is still a result so they should be shown.

Corrected

4.9 Why are some of the numbers in Table 4 written in bold?

Corrected

4.10 Figure 8 – 2009 was unusual because there was no seasonal cycle – can you comment on why this was?

When we reviewed the raw, L1 and L2 data, other period captured the summer drop but 2009 had an instrumental issue in summer. We commented in the manuscript.

*P21 L20:* In 2009 there is no clear seasonal cycle. There was an instrumental issue in summer while other period captured summer drop by observations.

4.11 I didn't fully understand the PSS analysis – I think that needs some more detail

To explain PSS analysis in detailed we revised the section 2.7.

**Section 2.7:** To identify and illustrate the potential source distributions for regional pollutions, we calculated the PSS using the trajectory statistics approach, which has often been applied to estimate the potential source areas of greenhouse gases (Reimann et al., 2004, 2008; Li et al., 2017). The trajectory statistics approach was introduced first by Seibert et al. (1994). The underlying assumption of the method is that elevated concentrations at an observations site are proportionally related to the air mass residence time on a specific grid cell over which observed air mass has been passing. Thus, this method simply calculates the airmass residence time weighted mean concentrations for target compounds ($CH_4$ in this study) for the domain with $0.5 \times 0.5$ grids using the following formula (Eq. 2):

$$\overline{C_{(i,j)}} = \frac{\sum_{a=1}^{M} T_{(i,j,a)} C_a}{\sum_{a=1}^{M} T_{(i,j,a)}} \qquad \text{(Eq. 2)}$$

where $C_{(i, j)}$ represents the potential source strength of the grid cell i, j as a potential source region of the target compound ($CH_4$); a is the index of the trajectory; M is the total number of trajectories that passed through cell i, j; $C_a$ is the enhanced mole fraction (difference from background mole fractions mentioned in section 3.2 below) measured during the arrival of trajectory a; and $T_{(i,j,a)}$ is the residence time of trajectory a spent over grid cell i, j and which were calculated using the method described by Poirot and Wishinski (1986) as following formula (Eq.3)

$$T_{(i,j,a)} = \sum_{n=1}^{N} \frac{S_{(i,j,n,a)}}{V_{(n,a)}} \qquad \text{(Eq. 3)}$$

where $S_{(i,j,n,a)}$ is the length of that portion of the $n^{th}$ segment of the a back-trajectory which falls over grid cell i,j. $V_{(n,a)}$ is the average speed of the air parcel as it travels along the $n^{th}$ segment of the a back-trajectory. Backward trajectories were calculated using the HYSPLIT

model of the NOAA Air Resources Laboratory (ARL) using meteorological information from the Global Data Assimilation System (GDAS) model. For the trajectory reliability, we used only the 4-days (96h) backward trajectories at an altitude of 500 m above the mean sea level. To consider the influence of air masses on emissions at ground level, the air masses passing above the boundary layer height (BLH) were excluded. The BLH can was obtained from the HYSPLIT model. To exclude the influences of emission sources surrounding AMY stations, enhanced $CH_4$ data with wind speeds lower than 2 m·s$^{-1}$ were excluded from the PSS analysis. When we compare PSS analysis among AMY, JGS, and ULD using $CH_4xs$ data from 2016 to 2020, they showed the similar source regions while the coverage and $CH_4xs$ are different (Fig. S5.).

---

## Referee Report (RR1)

Comments on 'Measurement Report: Atmospheric CH4 at regional stations of the Korea Meteorological Administration/Global Atmospheric Watch Programme: measurement, characteristics and long term changes of its drivers' by H. Lee et al.

This revisions made have improved this paper. I suggest however that there is a careful grammar check before publication, in particular in the recently revised sections.

The Miller Tans analysis is an improvement. The source signatures are compared with the global averages from the Sherwood et al. database, but it would be better to compare with signatures for the regions, as isotopic signatures for the sources can vary regionally (e.g. Ganesan et al., Spatially Resolved Isotopic Source Signatures of Wetland Methane Emissions - Ganesan - 2018 - Geophysical Research Letters - Wiley Online Library).

Uncertainties in the Miller-Tans signatures were not given for the KL site (page 22, line 12).

Some of the graph axis labels do not have subscript 4 in $CH_4$ (figures 6 and 7).

Missing superscript 13 in $\delta C$ (page 3, line 6).

---

## Referee Report (RR2)

**Review's comments**

**Manuscript Number:** *acp-2022-600*

**Title:** Measurement report: atmospheric CH$_4$ at regional stations of the Korea Meteorological Administration/ Global Atmosphere Watch Programme: measurement, characteristics and long-term changes of its drivers
**Authors:** Lee H., Seo W.-I., Li S., Lee S, Kenea S, and Joo S.

  I have gone through the response to my comments and the revised manuscript and found the manuscript has been improved to some extent. However, I think that there remain some problems which should be resolved before acceptance. The authors should respond the following comments before publication.

1) I'm not sure what the sentence, "However, isotopic signature using $\delta^{13}$CH$_4$ explained that CH$_4$ sources are becoming mixture of not only biogenic but also fossil fuel sources in East Asia from 2006 to 2020", means. I believe that the CH$_4$ sources in East Asia included fossil fuel-related source also before 2006.

2) The added sentence in the revised manuscript, "… since generic water correction cannot be applied to each instrument in different environments" (page 4, line 21) is a little bit misleading. The previous studies including Rella et al. (2013) and Hazan et al. (2016) indicated that the generic manufacturer's water correction function does not necessarily provides appropriate values for wet air samples. However, as Rella et al. (2013) indicated, the preliminary reduction of water vapor of sample air less than 1% allow us to measure the CH$_4$ mole fraction within the level of GAW's compatibility goal ($\pm$2 ppb). So, I think it would be better for the authors to simply mention that the sample air is dried to reduce the humidity effect less than the detection limit instead of correcting the humidity effect based on the generic manufacturer's water correction function or independently determined water correction function.

3) In the revised manuscript, the authors mentioned "AMY started flask sampling for CH$_4$ and $\delta^{13}$C$_{(CH4)}$ in December 2013 with same method like TAP and these data were used only for characterization of CH$_4$ at AMY in section 3.5" (page 8, line

22-23). But there is no discussion based on the $\delta^{13}C_{(CH4)}$ data for AMY in this paper.

4) The authors should explain how to prepare the Miller-Tans plots shown in Fig. 9b, Fig. 6S, and Fig. 7s because most readers are not necessarily familiar with the plot. In particular, the basic equation for the Miller-Tans plot should be described in the manuscript. And then, the authors should clarify how to determine the background values of the $CH_4$ concentration and isotope ratio. Additionally, a proper reference should be cited in the manuscript for the Miller-Tans plot.

5) I'm not sure the discussion based on the regional isotopic signatures from the Miller-Tans plots in page 22 line 12-20. The isotopic values can give us the information about the mix of the isotopically different $CH_4$ sources of the regional emissions causing the excess $CH_4$ at TAP. Therefore, the temporal and spatial changes in the isotopic ratio probably reflect the changes in the mix of the sources. But there is no information about emission strength in the isotopic ratio for the regional emissions. Therefore, I think it is difficult to discuss the variation in the $CH_4$ growth rate from the result based on the isotopic analysis in this study. For example, I cannot understand why the authors conclude "This suggested that $CH_4$ growth rate in East Asia was affected not only biogenic but also pyrogenic sources, unlike global." (page 22, line 12-13). If the authors want to discuss the cause of the trend variation in East Asia, the isotopic ratios for the base line should be discussed. But the authors also mentioned "The long-term trend at AMY was very similar to the global trend" (page 21, line 21-22). So, I think the trend of isotopic ratio at AYM is also similar to the global trend. The similar discussion about the trend in East Asia based on the isotopic analysis is also described in Section 4 (page 24, line 14-15).

6) I'm curious about whether there is seasonality in the source regions (CN, CS, and KL) affecting the observation at TAP. If there is seasonality, does it explain to some extent the differences in the isotopic ratios for the source regions shown in this study? For example, if the air masses are transported from CN to TAP during winter, it is expected that the isotopic signature would be relatively higher because the biogenic $CH_4$ emissions are reduced during winter.

7) I found several mistakes in English in the manuscript as some examples are shown in the followings:

Page 1, line 20-21: "… growth rate increased 3.3 ppb yr$^{-1}$ …" → " … growth rate increased by 3.3 ppb yr$^{-1}$ …"

Page 1, line 30: "…compared with that of other long-lived greenhouse gases…"
→ " …compared with those of other long-lived greenhouse gases…

Page 7, line 18: "… days moving median of of HA…"

Page 10, line 2: "The BLH can was obtained …"

Page 22, line 13: "… was affected not only biogenic but also pyrogenic…"
→ "…was affected by not only biogenic but also pyrogenic …"

Therefore, I think that it should be better to check the language in general.

8) Added sentences for Eq. 1 (page 6, line 7-9) are rather confusing. The Eq. (1) simply expresses the dilution effect caused by the water vapor ($H_{act}$). But I cannot understand what the sentences, "Here $H_{act}$ is the $H_2O$ difference between laboratory and working standards. $C_{dry}$ is the $CH_4$ mole fraction of laboratory standard while $C_{dilution}$ is the expressed $CH_4$ mole fraction when $H_{act}$ existed.", mean. Do the authors consider the water mole fraction in the laboratory standard is zero? If not, the exact relationship among the $H_2O$ differences between laboratory and working standards ($H^W$-$H^L$), the $CH_4$ mole fraction of standard with $H^L$ ($C_{dilution}^L$) and $H^W$ ($C_{dilution}^W$) is expressed by the following equations:

$C_{dilution}^W = C_{dilution}^L - 0.01(H^W$-$H^L) \cdot C_{dry}$

Or

$C_{dilution}^W / C_{dilution}^L = (1 - 0.01 \cdot H^W) / (1 - 0.01 \cdot H^L)$,

where $C_{dry}$ represents the dry $CH_4$ mole fraction of the standard.

However, I think above discussion is too detailed. In the manuscript, it would be better to simply examine the absolute $H_2O$ mole fraction of the standard gases and the dried air samples and discuss the dilution effects according to Eq. (1). I believe that the resultant changes in the $CH_4$ mole fractions caused by the $H_2O$ dilution effect are negligible.

9) Page 5, line 28: The URL https://gml.noaa.gov/ccl/service.html should be https://gml.noaa.gov/ccl/services.html.

10) Page 23, Fig. 9(a): It would be better to clarify the boundaries of the northern China (CN), southern China and Korea local (KL) in the map.

---

## Author Response (AR2)

**Author's responses to reviewer's comments follow. A copy of the reviewer comment is given (with comment 'number') followed by a response (blue font).**

**Response to referee 1**

1. General comment

These revisions made have improved this paper. I suggest however that there is a careful grammar check before publication, in particular in the recently revised sections.

Thank you for your comments. We finalized the grammar checks and tried to improve manuscript as following specific comment.

2. The Miller Tans analysis is an improvement. The source signatures are compared with the global averages from the Sherwood et al. database, but it would be better to compare with signatures for the regions, as isotopic signatures for the sources can vary regionally (e.g. Ganesan et al., Spatially Resolved Isotopic Source Signatures of Wetland Methane Emissions - Ganesan - 2018 - Geophysical Research Letters - Wiley Online Library).

Thank you for the suggestion. We added the sentence below.

P22 L6: Regionally $CH_4$ emissions from wetlands in Siberia are at -69.9±5.5‰ while Hong Kong was more enriched at -56.9±3.8‰ (Ganesan et al.,2018). In northern China, a high coal emission area, heavy $CH_4$ signal appears from -35‰ to -50‰ (Feinberg et al., 2018).

Feinberg et al.(2018): Isotopic source signatures: Impact of regional variability on the δ13CH4 trend and spatial distribution, Atmos. Environ. https://doi.org/10.1016/j.atmosenv.2017.11.037

3. Uncertainties in the Miller-Tans signatures were not given for the KL site (page 22, line 12).

Corrected.

P22 L14: . KL showed the quite constant values from -55±1.6 to -54±3.1‰

4. Some of the graph axis labels do not have subscript 4 in CH4 (figures 6 and 7).

Corrected.

5. Missing superscript 13 in δC (page 3, line 6)

Corrected.

**Author's responses to reviewer's comments follow. A copy of the reviewer comment is given (with comment 'number') followed by a response (blue font).**

**Response to referee 2**

1. General comment

I have gone through the response to my comments and the revised manuscript and found the manuscript has been improved to some extent. However, I think that there remain some problems which should be resolved before acceptance. The authors should respond the following comments before publication.

Thank you for your comments. We tried to resolve the pointed problem here.

2. I'm not sure what the sentence, "However, isotopic signature using δ 13CH4 explained that CH4 sources are becoming mixture of not only biogenic but also fossil fuel sources in East Asia from 2006 to 2020", means. I believe that the CH4 sources in East Asia included fossil fuel-related source also before 2006.

Agree. We revised the sentence below:

Abstract P1 L23-24: "However, $\delta^{13}CH_4$ indicate that the $CH_4$ trend in East Asia is derived from biogenic and fossil fuel sources from 2006 to 2020.."

3. The added sentence in the revised manuscript, "… since generic water correction cannot be applied to each instrument in different environments" (page 4, line 21) is a little bit misleading. The previous studies including Rella et al. (2013) and Hazan et al. (2016) indicated that the generic manufacturer's water correction function does not necessarily provides appropriate values for wet air samples. However, as Rella et al. (2013) indicated, the preliminary reduction of water vapor of sample air less than 1% allow us to measure the CH4

mole fraction within the level of GAW's compatibility goal (±2 ppb). So, I think it would be better for the authors to simply mention that the sample air is dried to reduce the humidity effect less than the detection limit instead of correcting the humidity effect based on the generic manufacturer's water correction function or independently determined water correction function.

Thank you for your comment. We should make clear that we don't correct our values from the humidity effects because we are using a drying system. We just consider these values as one of our uncertainty factors to show our data are not affected by humidity. And as reviewer mentioned this value is negligible. Therefore, we revised the sentence like below and hope this is acceptable.

P4 L20: This system dried the sampled air enough so that the bias from the humidity is negligible. This is described in section 2.3 and 3.1.

4. In the revised manuscript, the authors mentioned "AMY started flask sampling for CH4 and δ 13C(CH4) in December 2013 with same method like TAP and these data were used only for characterization of CH4 at AMY in section 3.5" (page 8, line 22-23). But there is no discussion based on the δ 13C(CH4) data for AMY in this paper.

There is a typo section 3.4.3 and added the data are used in Fig. S6. We revised the sentence.

P8 L23: .. and these data were used only for characterization of $CH_4$ at AMY in section 3.4.3.

5. The authors should explain how to prepare the Miller-Tans plots shown in Fig. 9b, Fig. 6S, and Fig. 7s because most readers are not necessarily familiar with the plot. In particular, the basic equation for the Miller-Tans plot should be described in the manuscript. And then, the authors

should clarify how to determine the background values of the CH4 concentration and isotope ratio. Additionally, a proper reference should be cited in the manuscript for the Miller-Tans plot.

We added explanation and reference as well.

P9 L1: To understand the regional source signature of excess $CH_4$ and to consider background atmospheric variations, Miller-Tans plot are used with flask sample data (Miller & Tans, 2003).

$$\delta_{obs}C_{obs}-\delta_{bg}C_{bg} = \delta s(C_{obs}-C_{bg})$$

Here, C and $\delta$ refer to $CH_4$ and $\delta^{13}C_{(CH4)}$ and the subscript bg, obs and s refer to background, observed and source values. Therefore, by plotting $\delta_{obs}C_{obs}-\delta_{bg}C_{bg}$ (y) against $C_{obs}-C_{bg}$ (x), $\delta s$ indicates the slope of the linear regression represents the source $\delta^{13}C_{(CH4)}$ signature. For background data, we downloaded data of $CH_4$ and $\delta^{13}C_{(CH4)}$, observed at Mauna Loa (https://gml.noaa.gov/dv/data, last access March 2022)

6. I'm not sure the discussion based on the regional isotopic signatures from the Miller-Tans plots in page 22 line 12-20. The isotopic values can give us the information about the mix of the isotopically different CH4 sources of the regional emissions causing the excess CH4 at TAP. Therefore, the temporal and spatial changes in the isotopic ratio probably reflect the changes in the mix of the sources. But there is no information about emission strength in the isotopic ratio for the regional emissions. Therefore, I think it is difficult to discuss the variation in the CH4 growth rate from the result based on the isotopic analysis in this study. For example, I cannot understand why the authors conclude "This suggested that CH4 growth rate in East Asia was affected not only biogenic but also pyrogenic sources, unlike global." (page 22, line

12-13). If the authors want to discuss the cause of the trend variation in East Asia, the isotopic ratios for the baseline should be discussed. But the authors also mentioned "The long-term trend at AMY was very similar to the global trend" (page 21, line 21-22). So, I think the trend of isotopic ratio at AYM is also similar to the global trend. The similar discussion about the trend in East Asia based on the isotopic analysis is also described in Section 4 (page 24, line14-15)

We understand what you mean but when air is collected by flasks, we consider it is well-mixed air. This means flask air samples reflects baseline airmass rather than the polluted local air. And the baseline also reflects the air not only in northern hemisphere but also in the region. Therefore, the levels can be different from global values, but growth rate can be similar to those of global (For example, -0.026‰/year for Mauna Loa and -0.027‰/year at AMY from 2006 to 2020). With this reason, we applied the Miller-Tans plots to figure out only regional changes as reviewer suggested previously and applied the cluster analysis. Through the Miller-Tans plot, we remove the global (or northern hemisphere) trend and variation of $CH_4$ and its isotope. This means our $\Delta$ values are regional values after removing background baseline of northern hemisphere (Mauna Loa). Also, through the cluster analysis, we can assess which regions are enhanced or depleted with isotope signature from 2006 to 2020.

We assure this approach is reasonable to explain of $CH_4$ regional growth rate base on measurement data.

7. I'm curious about whether there is seasonality in the source regions (CN, CS, and KL) affecting the observation at TAP. If there is seasonality, does it explain to some extent the differences in the isotopic ratios for the source regions shown in this study? For example, if the air masses are transported from CN to TAP during winter, it is expected that the isotopic signature would be relatively higher because the biogenic CH4 emissions are reduced during winter.

That is true. As reviewer mentioned, CN includes summer data with only 1.6% of total data while 95% for CS. On the other hand, CN does not always include winter data. 41% of total CN data are observed in winter while 57.4% for spring and autumn. Therefore, it is difficult to explain CN is representative of winter. In the section 3.4.3, we already described the AMY's Miller-Tans plot result in winter and summer. Regardless of season, the slope of Miller-Tans plot at AMY is -52.3±2.2‰ in winter and -53.7±0.7‰ in summer as indicating depleted δC values from 2016 to 2020 (P21 L7). This result might be different from our assumption that isotopic signature in winter should be higher than summer. When we implement cluster analysis (as described in section 2.6), we can confirm the data derived from KL or stagnated condition showed the depleted δC values (closed to biogenic source signature, from -50 to -61‰) even in winter. Therefore, we can understand that $CH_4$ signature slope represents the source region even though we can consider the seasonal characteristics when we applied the cluster analysis. Additionally in section 3.5 we tried to explain $CH_4$ signature slope changes in the same cluster sector during the long-term period. Therefore, it can be less important how much winter and summer affected on this analysis.

8. I found several mistakes in English in the manuscript as some examples are shown in the followings: Page 1, line 20-21: "… growth rate increased 3.3 ppb yr-1 …" → " … growth rate increased by 3.3 ppb yr-1 …" Page 1, line 30: "…compared with that of other long-lived greenhouse gases…" → " …compared with those of other long-lived greenhouse gases… Page 7, line 18: "… days moving median of of HA…" Page 10, line 2: "The BLH can was obtained …" Page 22, line 13: "… was affected not only biogenic but also pyrogenic…" → "…was affected by not only biogenic but also pyrogenic …"

Therefore, I think that it should be better to check the language in general.

Corrected and whole manuscript was screened by grammar checks.

9. Added sentences for Eq. 1 (page 6, line 7-9) are rather confusing. The Eq. (1) simply expresses the dilution effect caused by the water vapor (Hact). But I cannot understand what the sentences, "Here Hact is the H2O difference between laboratory and working standards. Cdry is the CH4 mole fraction of laboratory standard while Cdilution is the expressed CH4 mole fraction when Hact existed.", mean. Do the authors consider the water mole fraction in the laboratory standard is zero? If not, the exact relationship among the H2O differences between laboratory and working standards (HW-H L ), the CH4 mole fraction of standard with H L (CdilutionL ) and H W (CdilutionW) is expressed by the following equations: CdilutionW = CdilutionL − 0.01(HW-H L ) ·Cdry Or CdilutionW / CdilutionL = (1 − 0.01·H W) / (1 − 0.01·H L ), where Cdry represents the dry CH4 mole fraction of the standard. However, I think above discussion is too detailed. In the manuscript, it would be better to simply examine the absolute H2O mole fraction of the standard gases and the dried air samples and discuss the dilution effects according to Eq. (1). I believe that the resultant changes in the CH4 mole fractions caused by the H2O dilution effect are negligible.

Once again, we don't apply drying correction to our data because the air is dried enough with our drying system. Therefore any result will not be changed in the manuscript.

We were keen to follow the reviewer's comment as long as we can and finally revised the sentences like below. Hope this is well reflected as following reviewer's comment.

P6 L1: where $C$ is the $CH_4$ mole fraction and $H_{act}$ is the water mole fraction difference between laboratory and standard gases (in %). For example, 0.00054% $H_2O$ difference between two cylinders causes 0.01 ppb bias at 1800 ppb.

10. Page 5, line 28: The URL https://gml.noaa.gov/ccl/service.html should be https://gml.noaa.gov/ccl/services.html.

Corrected

11. Page 23, Fig. 9(a): It would be better to clarify the boundaries of the northern China (CN), southern China and Korea local (KL) in the map.

Corrected

---

## Author Response (AR3)

**Author's responses to editor's comments follow. A copy of the reviewer comment is given (with comment 'number') followed by a response (blue font).**

**Public justification (visible to the public if the article is accepted and published)**:

I do not agree with your response to reviewer #2 stating:

"We understand what you mean but when air is collected by flasks, we consider it is well-mixed air. This means flask air samples reflects baseline airmass rather than the polluted local air. And the baseline also reflects the air not only in northern hemisphere but also in the region. Therefore, the levels can be different from global values, but growth rate can be similar to those of global (...)"

Air collection by flasks can be affected by high levels of local pollution and air sampling does not per se result in baseline data. For this reason the background selection method described in section 2.4.2 is vital. Samples would only then well-mixed if sampling times are very long, depending on the location of sampling and the local meteorological conditions during the sampling.

I am in general fine with the manuscript now, but since this issue came up, I ask the authors to add information about the typical sample collection time in the method section. Although stated elsewhere in literature, I think this detail should be mentioned in the current text.

Thank you for your review and comments.

I agree your comments. Since we described the sampling method already "Samples were collected weekly between 1200 and 1800 (Korea Local Time), when boundary layer height (BLH) was maximum", we tried to explain why the flask sample data we used do not include local impacts.

L17 P8: Samples were collected weekly between 1200 and 1800 (Korea Local Time), when boundary layer height (BLH) was maximum to reduce local impacts.

L8 P9: Observed $\delta^{13}C_{(CH4)}$ are selected by the cluster analysis (section 2.6).

L17 P9: Among clusters, 25% of samples are derived from under the stagnated condition which might be affected by local pollutions. Therefore, we did not consider this sector.

---

## Author Response (AR4)

**Author's responses to editor's comments follow. A copy of the reviewer comment is given (with comment 'number') followed by a response (blue font).**

Please add a typical sample collection duration to L17 P8. Over what period does a typical sample integrate, i. e. how long does the pressurization of the canisters take? This is what I had asked for following the discussion about review#2, but I cannot find this information in the revised manuscript..

We added the description about flask sampling. We apologize our misunderstanding.

L17 P8: The pair of flask-air samples (2L each flask, borosilicate glass with Teflon O-ring sealed stopcocks) was flushed for 10 min at 5-6 L min$^{-1}$ then pressurized to 0.38 bar in less than 1 min using a semi-automated portable sampler.